## Response to Reviewers

We greatly appreciate the constructive comments from all reviewers and the opportunity we were given to improve our paper. Following your comments and advice, we revised our manuscript to include missing details and planned analysis to clarify unclear parts to make our study more sound and complete. Within the points, we also reviewed all the detailed comments from the reviewers with our best efforts. We refer to specific locations of each change using  Review X-XX  . You can click **X** ("on page **X**") to jump to the modified location. We are currently on revision R1.1, which has addressed some of the identified issues. Revision R1.2 is planned next and will focus on additional experiments, along with addressing all remaining clarifications mentioned in this revision. We anticipate requiring four weeks from now to complete R1.2. Following that, revision R1.3 will focus on conducting the ablation study and reorganizing the empirical results sections. We expect this phase to require an additional three weeks.

**Point 1: The contributions of the proposed approach in contrast to other works**

*Reviewer #P9yW: This work has similarities to (Leontiadis et al., 2021). The authors in (Leontiadis et al., 2021) present a framework that is capable of handling unlabeled data (self-distillation) similar to the proposed framework in addition to handling labeled data. It would be beneficial to cite and compare the self-distillation case of this work.*

We would like to thank the reviewer for suggesting this existing work. We have thoroughly reviewed this work and discussed the differences. In short, in this paper, they try to train their cache models using labels and as they have mentioned, it is semi-supervised. But we train the layers using only the trained models. Not using any labels during training the caches is our main contribution compared to existing work. We have added additional explanations in Section 2.3. The content marked with  Review 1-1  **on page 8** is the location of the change.

*Reviewer #uRWL: The revision should better clarify the novel contributions of the proposed approach and contrast it to the existing work.*
Our work's main novelty is about training the cache models only at inference time and without any labels. We will make this distinction with related work more explicit in the R1.2.

*Reviewer #uRWL: The reduction of computation is not significant compared with network compression methods (Table 2) [(Liu et al., 2019a),(Li et al., 2020b)]. In addition, the model size is increased and the memory usage could also be increased.*
In MetaPruning and DHP mentioned in the comment, the authors use search algorithms for channel pruning and they suggest compressed networks with lighter models, which lead to a lower accuracy but faster inference. The computation cost is really better than ours, in some certain cases. But our approach does not lose as much accuracy as them in most of the time, while showing faster inference. In fact in most cases either they offer better computation but have less accuracy, or better accuracy with higher latency. In addition, in our approach, computation can be tuned with the confidence parameter, which would be another advantage for us. We will ensure to highlight these observations more clearly in R1.2. Regarding the second part of the comment: Indeed, in certain scenarios, adding more cache layers and consequently increasing memory usage does not lead to significant improvements, but we demonstrate better accuracy using cache methods.

**Point 2: Latency/computational cost/memory comparison**

*Reviewer #P9yW: Providing a comparison of latency/computational cost of other techniques mentioned in the related works (such as inference optimization) would strengthen the novelty of the proposed framework.*
*Reviewer #P9yW: How does the proposed framework compare in terms of the runtime memory requirements? Making a comparison of this factor in addition to the latency and computational comparisons would provide a more wholistic (and a stronger) picture of the framework.*
Thanks for the suggestions. In the R1.2 revision we will add latency/ computational cost results for the

baseline techniques, on Resnets and CIFARs. 4.5.4 already explains the details of run-time memory consumption, but given your comment we will provide further details on memory consumption in comparison to other baseline in RQ4. Note that object detection can not be implemented using the related works, since they are optimized and developed only for classification methods.

### Point 3: Considering other application

*Reviewer #P9yW: The authors should consider demonstrating the value of the proposed framework using other applications, ones where a drop in accuracy can be tolerated, and even preferred for gaining computational/latency benefits.*
*Reviewer #uRWL: Experiments are not done on large scale datasets such as ImageNet.*
Thanks for the suggestions. Please note that even with our current case studies the drop in accuracy is minimal (less than 1% in the best cases). Nonetheless, we plan to add another case study of DLRM (Deep Learning Recommendation Model) on Criteo Ad Kaggle dataset in R1.2. We selected this case study for these reasons:

(a) Unlike the previous case we had in the paper, the recommendation systems in practice can tolerate accuracy decrease to achieve lower latency and higher response rate.

(b) The dataset is much larger than the two we used to have.

(c) The task is completely new so it helps in verifying the generalizability of the findings.

### Point 4: Ablation study

*Reviewer #P9yW: The optimum subset of these cached models (that work hand in hand) are then selected using a scoring mechanism. How does this work in comparison to training cached models together? Providing this information could help the reader understand the need for (and even appreciate) the proposed training technique.*
*Reviewer #P9yW: The search space for the cache models have been restricted to 2 convolution layers/2 fully connected layers (for the author's experiments). Would increasing this space by allowing deeper (3 to 5 layers) cache models allow early exits at earlier layers? If so, would this be computationally less expensive than the shallower cached models exiting at later layers of the backbone? An ablation study exploring these questions would be a good addition to the paper.*
*Reviewer #qwkm: Why is NAS necessary and important? Why using the same architecture for the cache model would not work?*
*Reviewer #uRWL: Ablation studies regarding the alternative choices for each component are suggested. The authors should at least provide better justifications for their design in Sec. 3.*
To find the best subset to use as our cache models and layers we benefit from the NAS approach. It does not guarantee to select the best model (it is reasonable because it only performs on limited models and data without complete training), we have designed an ablation study which can test both our methods and this scoring as well. In R1.3 revision we will add an ablation study as follows: We will focus on the design of cache layers and study three different configurations:

(a) Using NAS we will find the top N solutions and we see how the results are different from each other.

(b) Without NAS, we pick several random cache configurations and run it on all models and show how much decrease of performance we get compare to the NAS-optimized approach along with the statistical significance of the results.

(c) We also do a small sensitively analysis to see to what extent the choice of cache layers' numbers/type matter.

**Point 5: FLOPs and cache-enabled latency time calculation**

*Reviewer #P9yW: How are the cache-enabled FLOPs and cache-enabled latency time calculated in section 4.5.5 when utilizing larger batch sizes? Are the FLOPs of all cached models utilized by each of the inputs in the batch aggregated? Is the latency the largest time for any input in the batch?*
This is indeed a valuable point to highlight. Yes, the FLOPs of all our cached models, utilized by each input in the batch, are aggregated. Yes, latency is the largest time seen in the batch. The content marked with $\boxed{\text{Review 1-5}}$ **on page 23** is the location of the change.

**Point 6: Organizing and clarifying the sections**

*Reviewer #uRWL: Sec. 3 needs rigorous explanations to justify the design of each of the steps in the proposed approach.*
In a nutshell, in the first step, we analyze each selected model to find the candidate positions for cache layers. Layers with the outputs without any other layer's state involved are the candidates which can enable training possibility. Then using NAS, we evaluate all possible subsets we can put in cache layers. NAS would also score between architectures we have defined for our cache models. Next we train the identified optimal layers at inference time without any labels. Finally, we test our cache models. We modified titles and will add some more words to better explain this in R1.3 revision (3).
*Reviewer #uRWL: Sec. 4.5 is not well-organized. I would personally suggest having an individual section, namely empirical evaluation results. Since the authors have explicitly defined four research questions, I would suggest nesting results and findings under each research question.*
*Reviewer #qwkm: It is better that the Sec 4.4 and Sec 4.5 are merged. The explanation of the metrics occurs where it is used for the first time.*
In R1.3 revision, we will make two separate sub-sections in 4: 4.1 Evaluation Setting and 4.2 Evaluation Results, where each RQ will have its own subsection. Given this reorganization, we believe Sec 4.4 which explains metrics fall under design not results, and should go with Section 4 not 5. After adding one more application and ablation study, they will be modified.
*Reviewer #uRWL: Currently, Sec. 4.5.2 and 4.5.3 are in awkward positions. Finally, using one or two sentences to summarize the answers to each research question may improve the paper's readability.*
We will add summary boxes per question in R1.2.

**Point 7: Revising vague sections and formulations**

*Reviewer #qwkm: After reading Section 3.3, it is still difficult to understand how confidence calibration is done. It is easy to understand what confidence calibration is. But it is not explained why the confidence of a trained cache model could be improved by using the validation set. This part is quite vague.*
In short, we employ a threshold to gauge the confidence level of the cache based on its model's output. Each cache functions as a classifier, and its output, during validation or testing, carries a confidence value. We will make further efforts to clarify this aspect in R1.2 revision.
*Reviewer #qwkm: In the last but second paragraph of Sec. 3.3, notations are not explained well. What do T and X and n mean? Why is the accuracy drop decreases with the increase of the number of layers?*
It seems like we had a typo and X is wrongly mentioned in the paper and should be T which is the tolerance for drop in the final accuracy. n is the 1-based index of the cache model in the setup. The formulation has been fixed in this revision. Regarding the second question, utilizing additional cache layers may cause earlier detection of a class with a higher probability (with using an appropriate confidence for cache layers) before moving to the next layers, so the total accuracy would increase with the cost of memory consumption. $\boxed{\text{Review 1-7-1}}$ **on page 13** is the location of the change.
*Reviewer #qwkm: In Section 3.4, it is not clear how are the subsets S decided. Once the*

*subsets are generated, is it true that the subset with the best score in Equation 4 is selected?*
Exactly. After running our algorithm, we will select the optimal subset for use in our cache layers.
Review 1-7-2 **on page 11** tries to clarify it.
*Reviewer #qwkm: Table 3 and Table 4 should be combined with the metric of accuracy drop. Considering the FLOPs or memory usage solely without mentioning accuracy drop makes no sense.*
You are correct. We will add accuracy drops to the tables as well to make the comparison easier for the readers in R1.2.
*Reviewer #qwkm: Are the architecture searching and training of the cache model conducted simultaneously?*
No. We initially select the best model through NAS, and subsequently, we conduct our experiments and cache training in the next stage. We will make it more clear in the revised paper in R1.2.
*Reviewer #qwkm: Please explain in detail the following: More specifically, even if a cache model shows promising hit rate and accuracy in individual evaluation, its performance in the deployment can be affected due to the previous cache hits made by the earlier cache models (connected to shallower layers in the backbone)*
We employ cache layers in series, meaning if a cache layer performs poorly in the early stages of our network and triggers an incorrect early exit, then a superior cache model positioned in the later stages might not be utilized. Consequently, this scenario could negatively impact the overall performance. For example, assume that we conduct an image for inference and a cache layer hits it soon with a wrong prediction. Then later layers will not happen to observe the input to decide about another early exit and the whole performance will decrease. So, the individual cache model's accuracy/hit rate don't guarantee the overall good results since the final results depend on the earlier cache model's results as well. We plan to include our detailed explanation in R1.2.

**Point 8: Writing issues**

*Reviewer #P9yW: Some typographical changes.*
*Reviewer #qwkm: The paper is not well written, the writing should be improved. A lot of details are not well explained.*
We make sure we will proof-read the revised revisions again more carefully.

# Improving Efficiency of Neural Image Classification and Object Detection Systems using Automated Layer Caching

**Anonymous authors**

## Abstract

Deep Neural Networks (DNNs) have become an essential component in many application domains including web-based services. A variety of these services require high throughput and (close to) real-time features, for instance, to respond or react to users' requests or to process a stream of incoming data on time. However, the trend in DNN design is toward larger models with many layers and parameters to achieve more accurate results. Although these models are often pre-trained, the computational complexity in such large models can still be relatively significant, hindering low inference latency. In this paper, we propose an end-to-end automated caching solution to improve the performance of DNN-based services in terms of their computational complexity and inference latency. Our method adopts the ideas of self-distillation of DNN models and early-exits. The proposed solution is an automated online layer caching mechanism that allows early-exiting of a large model during inference time if the cache model in one of the early-exits is confident enough for final prediction. One of the main contributions of this paper is that we have implemented the idea as an online caching, meaning that the cache models do not need access to training data and perform solely based on the incoming data at run-time, making it suitable for applications using pre-trained models. Our experiments results on two downstream tasks (image classification and object detection) show that, on average, caching can reduce computational complexity of these services up to 58% (in terms of FLOPs count) and improve their inference latency up to 46% with low to zero reduction in accuracy. Our approach also outperforms existing approaches, particularly when being applied on complex models and larger datasets. It achieves a remarkable 51.6% and 30.4% reduction in latency, surpassing the Gati and BranchyNet methods for CIFAR100-Resnet50. This enhancement is accompanied by 2.92% and 0.87% increase in mean accuracy, further highlighting the superiority of our approach in demanding scenarios.

## 1 Introduction

Deep Neural Networks (DNNs) are incorporated in real-world applications used by a broad spectrum of industry sectors including healthcare (Shorten et al., 2021; Fink et al., 2020), finance (Huang et al., 2020; Culkin, 2017), self-driving vehicles (Swinney & Woods, 2021), and cybersecurity (Ferrag et al., 2020). These applications utilize DNNs in various fields such as computer vision (Hassaballah & Awad, 2020; Swinney & Woods, 2021), audio signal processing (Arakawa et al., 2019; Tashev & Mirsamadi, 2017),and natural language processing (Otter et al., 2021). Many services in large companies such as Google and Amazon have DNN-based back-end software (e.g., Google Lens and Amazon Rekognition) with tremendous volume of queries per second. For instance, Google processes over 99,000 searches every second (Mohsin, 2022) and spends a substantial amount of computation power and time at their models' run-time (Xiang & Kim, 2019). These services are often time-sensitive, resource-intensive, and require high availability and reliability.

Now the question is how fast the current state-of-the-art (SOTA) DNN models are at inference time and to what extent they can provide low latency responses to queries. The SOTA model depends on the application domain and the problem at hand. However, the trend in DNN design is indeed toward pre-trained large-scale

models due to their reduced training cost (only fine-tuning) while providing dominating results (since they are huge models trained on an extensive dataset).

One of the downsides of large-scale models (pre-trained or not) is their high inference latency. Although the inference latency is usually negligible per instance, as discussed, a relatively slow inference can jeopardize a service's performance in terms of throughput when the QPS is high.

In general, in a DNN-based software development and deployment pipeline, the inference stage is part of the so called "model serving" process, which enables the model to serve inference requests or jobs (Xiang & Kim, 2019) by directly loading the model in the process or by employing serving frameworks such as TensorFlow Serving (Olston et al., 2017) or Clipper (Crankshaw et al., 2017).

The inference phase is an expensive stage in a deep neural model's life cycle in terms of time and computation costs (Desislavov et al., 2021). Therefore, efforts towards decreasing the inference cost in production have increased rapidly throughout the past few years.

From the system engineering perspective, caching is a standard practice to improve software systems performance, which helps avoid redundant computations. Caching is the process of storing recently observed information to be reused when needed in the future, instead of re-computation (Wessels, 2001; Maddah-Ali & Niesen, 2014). Caching is usually orthogonal to the underlying procedure, meaning that it is applied by observing the inputs and outputs of the target procedure and does not engage with the internal computations of the cached function.

Caching effectiveness is best observed when the cached procedure often receives duplicated inputs while in a similar internal state—for instance, accessing a particular memory block, loading a web page, or fetching the books listed in a specific category in a library database. It is also possible to adopt a standard caching approach with DNNs (e.g., some work cache a DNN's output solely based on its input values (Crankshaw et al., 2017)). However, it would most likely provide a meager improvement due to the high dimension and size of the data (such as images, audios, texts) and low duplication among the requests.

However, due to the feature extracting nature of the deep neural networks, we can expect the inputs with similar outputs (e.g., images of the same person or the same object) to have a pattern in the intermediate layers' activation values. Therefore, we exploit the opportunity to cache a DNN's output based on the intermediate layers' activation values. This way, **we can cache the results not by looking at the raw inputs but by looking at their extracted features in the intermediate layers within the model's forward-pass**.

The intermediate layers often have even higher dimensions than the input data. Therefore, we use shallow classifiers (Kaya et al., 2019) to replace the classic cache storing and look-up procedures. A shallow classifier is a supplementary model attached to an intermediate layer in the base model that uses the intermediate layer's activation values to infer a prediction. In the caching method, training a shallow classifier on a set of samples mimics the procedure of storing those samples in a cache storage, and inferring for a new sample using the shallow classifier mimics the look-up procedure.

In this paper, we propose caching the predictions made by off-the-shelf classification models using shallow classifiers trained using the samples and information collected at inference time. We first evaluate the rationality of our method in our first research question by measuring how it affects the final accuracy of the given base models and assessing the effectiveness of the parameters we introduce (tolerance and confidence thresholds) as a knob to control the caching certainty. We further evaluate the method in terms of computational complexity and inference latency improvements in the second and third research questions. We measure this improvements by comparing the FLOPs count, memory consumption, and inference latency for the original model vs. the cache-enabled version that we build throughout this experiment. We observed up to 58% reduction in FLOPs, up to 46% acceleration in inference latency while inferring on CPU and up to 18% on GPU, with less than 2% drop in accuracy. Our method demonstrates remarkable performance across a range of models and datasets. For the simplest model and dataset, CIFAR10-resnet50, it offers a substantial 52.2% and 32.4% reduction in latency, with only a minor 6.1% and 0.9% drop in accuracy compared to BranchyNet and Gati methods, respectively. Furthermore, in the case of the more complex CIFAR100-Resnet50 model and dataset, our method achieves a significant 51.6% and 30.4% latency reduction

while concurrently enhancing mean accuracy by 2.92% and 0.87% when compared to Gati and BranchyNet methods.

In summary the contributions of this paper are:

- Proposing a caching method for the predictions made by off-the-shelf image classifiers and object detection models, which only uses unlabelled samples collected at inference time.

- Automating the process of designing the supplementary models used for caching and tuning their parameters used for determining the cache hits, by AutoML methods.

- Empirically evaluating the proposed caching method using 5 publicly available off-the-shelf models on 4 datasets, in terms of computational complexity and inference time reduction.

In the rest of the paper, we discuss the background and related works in section 2, details of the method in section 3, design and evaluation of the study in section 4, and we conclude the discussions in section 5.

## 2 Related Works

In this section, we briefly review the topics related to the model inference optimization problem. Following this, we introduce the techniques used to build the caching procedure.

### 2.1 Inference Optimization

There are two perspectives addressing the model inference optimization problem. The first perspective is interested in optimizing the model deployment platform and covers a broad range of optimization targets (Yu et al., 2021). These studies often target the deployment environments in resource-constrained edge devices (Liu et al., 2021; Zhao et al., 2018) or resourceful cloud-based devices (Li et al., 2020a). Others focus on hardware-specific optimizations (Zhu & Jiang, 2018) and inference job scheduling (Wu et al., 2020).

The second perspective is focused on minimizing the model's inference compute requirements by compressing the model. Among model compression techniques, model pruning (Han et al., 2015; Zhang et al., 2018; Liu et al., 2019b), model quantization (Courbariaux et al., 2015; Rastegari et al., 2016; Nagel et al., 2019), and model distillation (Bucila et al., 2006; Polino et al., 2018; Hinton et al., 2015) are being extensively used. These ideas alleviate the model's computational complexity by pruning the weights, computing the floating-point calculations at lower precision, and distilling the knowledge from a teacher (more complex) model into a student (less complex) model, respectively. These techniques modify the original model and often cause a fixed amount of loss in the test accuracy.

As we mentioned in the motivation, inference acceleration is crucial for sensitive real-time approaches like autonomous vehicles. Most of the methods focus on partitioning and offloading calculations to the edge ((Mohammed et al., 2020)). However, achieving faster decisions for a vehicle to detect a pedestrian requires a more immediate reaction than outsourcing data to the edge. We have applied our method to state-of-the-art object detection approaches, such as Mask R-CNN ((He et al., 2017)) using popular urban datasets, demonstrating a significant improvement even when utilizing unlabeled data.

### 2.2 Early-Exits in DNNs

"Early-exit" generally refers to an alternative path in a DNN model which can be taken by a sample instead of proceeding to the next layers in the model. Many previous works have used the early-exit concept for different purposes (Xiao et al., 2021; Scardapane et al., 2020; Matsubara et al., 2022; Haseena Rahmath et al., 2023). (Panda et al., 2016) is one of the first and brilliant works in this area. They try to terminate classification by cascading a linear network of output neurons for each convolutional layer and monitoring the output of the linear network to decide about the difficulty of input instances and conditionally activate the deeper layers of the network. But they have not mentioned anything about inference time and accuracy/time trade-off issue. BranchyNet ((Teerapittayanon et al., 2016)), (Pacheco et al., 2021) and (Ebrahimi et al.,

2022) also utilize their old observation that features learned at an early layer of a network to make an early-exit. However, they require labeled data to train their models, rendering them unsuitable for use with unlabeled data. Shallow Deep Networks (SDN) (Kaya et al., 2019) points out the "overthinking" problem in deep neural networks. "Overthinking" refers to the models spending a fixed amount of computational resources for any query sample, regardless of their complexity (i.e., how deep the neural network should be to infer the correct prediction for the sample). Their research proposes attaching shallow classifiers to the intermediate layers in the model to form the early-exits. Each shallow classifier in SDN provides a prediction based on the values of the intermediate layer to which it is attached.

On the other hand, (Xiao et al., 2021) incorporates the shallow classifiers to obtain multiple predictions for each sample. In their method, they use early-exits as an ensemble of models to increase the base model's accuracy.

The functionality of the shallow classifiers in our proposed method is similar to SDN. However, the SDN method trains the shallow classifier using the ground truth data in the training set and overlooks the available knowledge in the original model. This constraint renders the proposed method useless when using a pre-trained model without access to the original training data, which is commonly the case for practitioners.

## 2.3 DNN Distillation and Self-distillation

Among machine learning tasks, the classification category is one of the significant use cases where DNNs have been successful in recent years. Classification is applied to a broad range of data such as image (Bharadi et al., 2017; Xia et al., 2021), text (Varghese et al., 2020), audio (Lee et al., 2009), and time-series (Zheng et al., 2014) classification.

Knowledge distillation(KD) (Bucila et al., 2006; Polino et al., 2018; Hinton et al., 2015) is a model compression method that trains a relatively small (less complex) model known as the student to mimic the behavior of a larger (more complex) model known as the teacher. Classification models usually provide a probability distribution (PD) representing the probability of the input belonging to each class. KD trains the student model to provide similar PDs (i.e., soft labels) to the teacher model rather than training it with just a class label for each sample (i.e., hard labels). KD uses specialized loss functions in the training process, such as Kullback-Leibler Divergence (Joyce, 2011) to measure how one PD is different from another.

KD usually is a 2-step process consisting of training a large complex model to achieve high accuracy and distilling its knowledge into a smaller model. An essential challenge in KD is choosing the right teacher and student models. Self-distillation (Zhang et al., 2021) addresses this challenge by introducing a single-step method to train the teacher model along with multiple shallow classifiers. Each shallow classifier in self-distillation is a candidate student model which is trained by distilling the knowledge from one or more of the deeper classifiers. In contrast to SDN, self-distillation utilizes knowledge distillation to train the shallow classifiers. However, it still trains the base model from scratch along with the shallow classifiers, using the original training set. This training procedure conflicts with our objectives in both aspects. Specifically, we use a pre-trained model and keep it unchanged throughout the experiment and only use inference data to train the shallow classifiers.

**Review 1-1** **[[ In (Leontiadis et al., 2021) the authors present a method for enhancing CNN efficiency through early exits. It leverages supervision, self-supervision, and self-distillation for on-device personalization, using both labeled and unlabeled data. This allows for dynamic adaptation with varying data availability, focusing on training enhancements. Our work is different by not altering the main model, but instead utilizing cache layers updated solely during the inference time. This is done to improve latency, without the need for labeled data during these updates, which offers a modular solution with minimal modifications to existing systems. ]]**

Our work modifies and puts the presented methods in SDN and self-distillation in the context of caching the final predictions of pre-trained DNN models. The method trains the shallow classifiers using only the unlabelled samples collected at run-time and measures the improvement in inference compute costs achieved by the early-exits throughout the forward-passes.

### 2.4 DNN Prediction Caching

Clipper (Crankshaw et al., 2017) is a serving framework that incorporates caching DNNs predictions based on their inputs. Freeze Inference (Kumar et al., 2019) investigates the use of traditional ML models such as K-NN and K-Means to predict based on intermediate layers' values. They show that the size and computation complexity of those ML models grows proportionally with the number of available samples and their computational overheads by far exceed any improvement. In Learned Caches, (Balasubramanian et al., 2021) extend the Freeze Inference by replacing the ML models with a pair of DNN models. A predictor model predicting the outputs and a binary classifier predicting whether the output should be used as the final prediction. Their method uses the ground truth data in the process of training the predictor and selector models. In contrast, our method 1) only uses unlabelled inference data, 2) automates the process of cache-enabling, 3) uses a confidence-based cache hit determination, 4) handles batch processing by batch shrinking.

## 3 Methodology

In this section, we explain the method to convert a pre-trained deep neural model (which we call the backbone) to its extended version with our caching method (called cache-enabled model). The caching method adds one or more early-exit paths to the backbone, controlled by the shallow classifiers (which we call the cache models), allowing the model to infer a decision faster at run-time for some test data samples (cache hits). Faster decision for a portion of queries will result in a reduced mean response time.

"Cache model" is a supplementary model that we attach to an intermediate layer in the backbone, and given the layer's values provides a prediction (along with a confidence value) for the backbone's output. Just a reminder that as our principal motivation, we assume that the original training data is unavailable for the user, as is the case for most large-scale pre-trained models used in practice. Therefore, in the rest of the paper, unless we explicitly mention it, the terms dataset, training set, validation set, and test set all refer to the whole available data at run-time or a respective subset.

Our procedure for cache-enabling a pre-trained model is chiefly derived from the self-distillation method (Zhang et al., 2021). However, we adopt the method to cache-enable pre-trained models using only their recorded outputs, without access to the ground truth (GT) labels.

A step-by-step guide on cache-enabling an off-the-shelf pre-trained model from a user perspective contains the following steps:

1. Identify the candidate layers to be cached

2. Build a cache model for each candidate

3. Assign confidence thresholds to the built models for determining the cache hits

4. Evaluate and optimize the cache-enabled model

5. Early-Exit Optimization Implementation

6. Update and maintenance

In the following subsections, we further discuss the procedure and design decisions in each step outlined above.

### 3.1 Identifying candidate layers

Choosing which layers to cache is the first step toward cache-enabling a model. A candidate layer is a layer that we will examine its values correlation to the final predictions by training a cache model based on them. One can simply list all the layers in the backbone as candidates. However, since we launch a search for a cache model per candidate layer in the next step, we suggest narrowing the list by filtering out some layers with the following criteria:

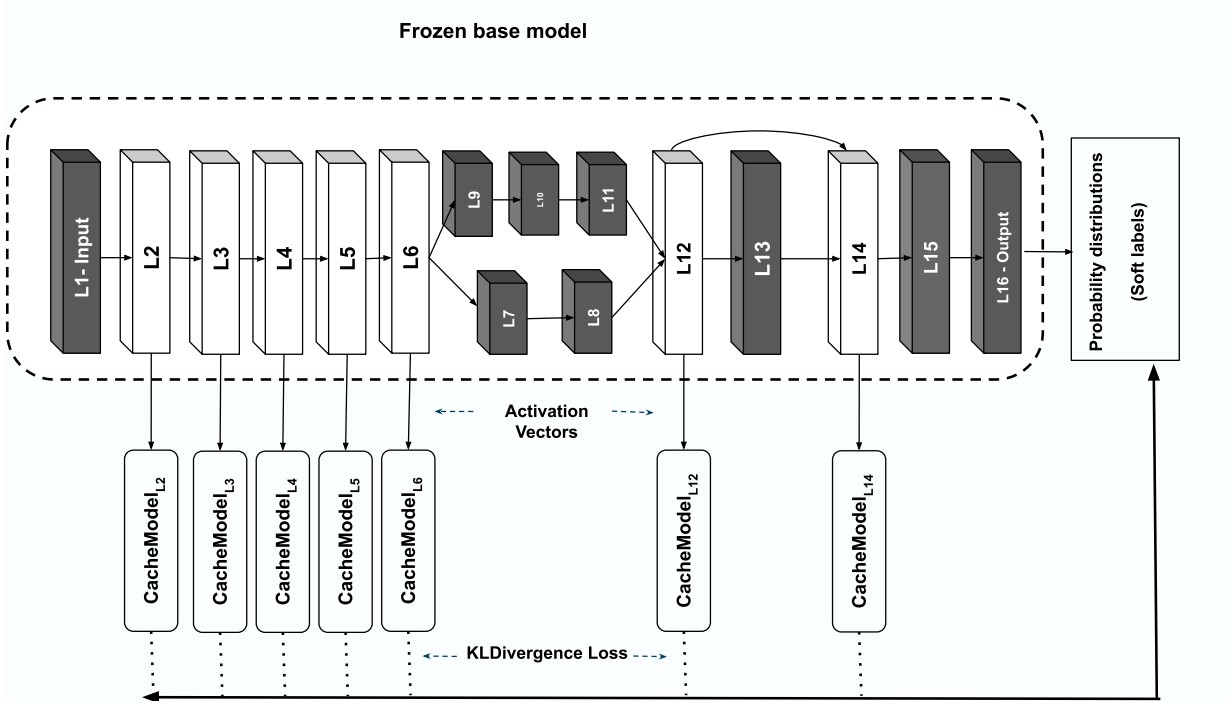

Figure 1: Cache-enabling procedure, candidate layers, and data paths.

- Some layers are disabled at inference time, such as dropouts and batch normalizations. These layers do not modify their input values at inference time. Therefore, we cross them off the candidates list.

- A few last layers in the model (close to the output layer, such as `L15` in Figure 1) might not be valuable candidates for caching since the remaining layers might not have heavy computations to reach the output.

- DNN models usually are composed of multiple components (i.e. first-level modules) consisting of multiple layers such as multiple residual blocks in ResNet models He et al. (2016)). We narrow down the search space to the outputs layers in those components.

- We only consider the layers which, given their activation values, the backbone's output is uniquely determined without any other layer's state involved (i.e., the backbone's output is a function of the layer's output). In other words, a layer with other layers or connections in parallel (such as `L7-L11` and `L13` in the Figure 1) is not suitable for caching since the backbone's output does not solely depend on the layer's output.

Having the initial set of the candidate layers, we next build and associate a cache model to each one.

## 3.2 Building cache models

Building a cache model to be associated with an intermediate layer in the backbone consists of finding a suitable architecture for the cache model and training the model with that architecture. The details of the architecture search (search space, search method, and evaluation method) and the training procedure (training data extraction and the loss function) are discussed in the following two subsections.

### 3.2.1 Cache models architecture

A cache model can have an architecture with any size in depth and breadth, as long as it provides more computational improvement than its overhead. In other words, it must have substantially less complexity

(i.e., number of parameters and connections) than the rest of the layers in the backbone that come after the corresponding intermediate layer. The search space for such models would contain architectures with different numbers and types of layers (e.g., a stack of dense and/or convolution layers). Nevertheless, all the models in the search space must output a PD identical to the backbone's output in terms of size (i.e., the number of classes) and activation (e.g., SoftMax or LogSoftMax).

In our experiments, the search space consists of architectures with a stack of (up to 2) convolution layers followed by another stack of (up to 2) linear layers, with multiple choices of kernel and stride sizes for the convolutions and neuron counts for the linear layers. However, users can modify or expand the search space according to their specific needs and budget.

The objective of the search is to find a minimal architecture that converges and predicts the backbone's output with acceptable accuracy. Note that any accuracy given by a cache model (better than random) can be helpful as we will have a proper selection mechanism later in the process to only use the cache predictions that are (most likely) correct, and also to discard the cache models yielding low computational improvement.

The user can conduct the search by empirically sampling through the search space or by using a automated Neural Architecture Search (NAS) tool such as Auto-Keras (Jin et al., 2019), Auto-PyTorch (Zimmer et al., 2021), Neural Network Intelligence (NNI) (Microsoft, 2022), or NASLib (Ruchte et al., 2020). However, we used NNI to conduct the search and customized the evaluation process to account for the models' accuracy and their computational complexity. We have used the floating point operations (FLOPs) count as the estimation for the models' computational complexity in this stage.

Several factors influence a cache model's architecture for a given intermediate layer. These factors include the target intermediate layer's dimensions, its position in the backbone, and the dataset specifications such as its number of target classes. For instance, the first cache models in CIFAR100-Resnet50 and in CIFAR10-Resnet18 experiments (shown as cache1 in Figure 6) have the same input size, but since CIFAR100 has more target classes, it reasonably requires a cache model with more learning capacity. Therefore, using NAS to design the cache models helps automate the process and alleviate deep learning expert supervision in designing the cache models.

| Review 1-7-2 | **[[ NAS tries to minimize the total accuracy no more than the tolerance so for all possible subsets, there will be a maximum range. This method does not guarantee the best score but is a good solution. Subsets with the best score will be selected as our cache system which will be added inside the model to be trained individually by the predictions and perform early exits. ]]**

Regardless of the search method, evaluating a nominated architecture requires training a model with the given architecture which we discuss the procedure in the next section. Moreover, since the search space is limited in depth, it is possible that for some intermediate layers, neither of the cache models converge (i.e., the model provides nearly random results). In such cases, we discard the candidate layer as non-suitable for caching.

### 3.2.2 Training a cache model

Figure (1) illustrates a cache-enabled model's schema consisting of the backbone (the dashed box) and the associated cache models. A cache model's objective is to predict the output of the backbone model, given the corresponding intermediate layer's output, per input sample.

Similar to the backbone, cache models are classification models. However, their inputs are the activation values in the intermediate layers. As suggested in self-distillation (Zhang et al., 2021), training a cache model is essentially similar to distilling the knowledge from the backbone (final classifier) into the cache model.

Therefore, to distill the knowledge from the backbone into the cache models, we need a medial dataset (MD) based on the collected inference data (ID). The medial dataset for training a cache model associated to an intermediate layer L in the backbone B consists of the set of activation values in the layer L and the PDs given by B per samples in the given ID, formally annotated as below:

$$MD_L = [i \in ID :< B_L(i), B(i) >] \tag{1}$$

where:

$MD_L$   : Medial dataset for the cache model associated with the layer `L`
`ID`   : The collected inference data consisting of unlabelled samples
$B_L$`(i)`: Activation values in layer `L` given the sample `i` to the backbone `B`
`B(i)` : The backbone's PD output for the sample `i`

Note that the labels in MDs are the backbone's outputs and not the GT labels, as we assumed the GT labels to be unavailable. We split the $MD_L$ into three splits ($MD_L^{Train}$, $MD_L^{Val}$, $MD_L^{Test}$) and use them respectively similar to the common deep learning training and test practices.

Similar to distillation method (Hinton et al., 2015), we use Kullback–Leibler Divergence (KLDiv) (Joyce, 2011) loss function in the training procedure. KLDiv measures how different are the two given PDs. Thus, minimizing the KLDiv loss value over $MD_L^{Train}$ trains the cache model to estimate the prediction of the backbone ($B(i)$).

Unlike self-distillation where (Zhang et al., 2021) train the backbone and the shallow classifiers simultaneously, in our method, while training a cache model, it is crucial to freeze the rest of the model including the backbone and the other cache models (if any) in the collection, to ensure the training process does not modify any parameter not belonging to the current cache model.

### 3.3  Assigning confidence threshold

The probability value associated to the predicted class (the one with the highest probability) is known as the model's confidence in the prediction. The cache model's prediction confidence for a particular input will indicate whether we stick with that prediction (cache hit) or we proceed with the rest of the backbone to the next — or probably final — exit (cache miss).

Confidence calibration means enhancing the model to provide an accurate confidence. In other words, a well-calibrated model's confidence accurately represents the likelihood for that prediction to be correct(Guo et al. (2017)). An over-confident cache model will lead the model to prematurely exit for some samples based on incorrect predictions, while an under-confident cache model will bear a low cache hit rate. Therefore, after building a cache model, we also calibrate its confidence using $MD_L^{Val}$ to better distinguish the predictions more likely to be correct. Several confidence calibration methods are discussed in (Guo et al., 2017), among which the temperature scaling (in the output layer) has shown to be practical and easy to implement.

Having the model calibrated, we next assign a confidence threshold value to the model which will be used at inference time to determine the cache hits and misses. When a cache model identifies a cache hit, its prediction is considered to be the final prediction. However, when needed for validation and test purposes, we obtain the predictions from the cache model and the backbone.

Table 1: Cache prediction confusion matrix, C: Cached predicted class, B: Backbone's predicted class, GT: Ground Truth label

| Category | B = C | B = GT | C = GT |
|---|---|---|---|
| $BC$ | ✓ | ✓ | ✓ |
| $\overline{BC}$ | ✓ | X | X |
| $B\overline{C}$ | X | ✓ | X |
| $\overline{B}C$ | X | X | ✓ |
| $\overline{B}\ \overline{C}$ | X | X | X |

A cache model's prediction (C) for an input to the backbone falls into one of the 5 correctness categories listed in table 1 with respect to the ground truth labels (GT) and the backbone's prediction (B) for the input.

Among the cases where the cache model and the backbone disagree, the $B\overline{C}$ predictions negatively affect the final accuracy and on the other hand, the $\overline{B}C$ predictions positively affect the final accuracy. The Equation 2 formulates a cache model's actual effect on the final accuracy.

$$F_\Delta(\theta) = \overline{B}C_\Delta(\theta) - B\overline{C}_\Delta(\theta) \tag{2}$$

Where:

$\Delta$ : The cache model
$F_\Delta$ : The actual accuracy effect $\Delta$ causes given $\theta$ as threshold
$B\overline{C}_\Delta$ : Ratio of $B\overline{C}$ predictions by $\Delta$ given $\theta$ as threshold
$\overline{B}C_\Delta$ : Ratio of $\overline{B}C$ predictions by $\Delta$ given $\theta$ as threshold

However, since we use the unlabelled inference data to form the MDs, we can only estimate an upper bound for the cache model's effect in the final accuracy. The estimation assumes that an incorrect cache would always lead to an incorrect classification for the sample ($\overline{B}C$). We estimate the change in the accuracy upper bound a cache model causes given a certain confidence threshold, by its hit rate and cache accuracy:

$$F_\Delta(\theta) \leq HR_\Delta(\theta) \times (1 - CA_\Delta(\theta)) \tag{3}$$

Where

$\Delta$ : The cache model
$F_\Delta$ : The expected accuracy drop $\Delta$ causes given $\theta$ as threshold
$HR_\Delta$ : Hit rate provided by $\Delta$ given $\theta$ as threshold
$CA_\Delta$ : Cache accuracy provided by $\Delta$ given $\theta$ as threshold

Given the tolerance $T$ for drop in final accuracy, we assign a confidence threshold to each cache model that yields no more than $T/2^n\%$ expected accuracy drop on $MD_L^{Val}$ according to the Equation 3, where n is the 1-based index of the cache model in the setup.

Review 1-7-1 [[ **Utilizing additional cache layers may cause earlier detection of a class with a higher probability (with using an appropriate confidence for cache layers) before moving to the next layers, so the total accuracy would increase with the cost of memory consumption.** ]] It is important to note that there are alternative methods to distribute the accuracy drop budget among the cache models. For instance, one can equally distribute the budget. However, as we show in the evaluations later in section 4.5.1, we find it reasonable to assign more budget to the cache models shallower positions in the backbone.

### 3.4 Evaluation and optimization of the cache-enabled model

So far, we have a set of cached layers and their corresponding cache models ready for deployment. Algorithm 1 demonstrates a Python-style pseudo implementation of cache-enabled model inference process. When the cache-enabled model receives a batch of samples, it proceeds layer-by-layer similar to the standard forward-pass. Once a cached layer's activation values are available, it will pass the values to the corresponding cache model and obtains an early prediction with a confidence value per sample in the batch. For each sample, if the corresponding confidence value exceeds the specified threshold, we consider it a cache hit. Hence, we have the final prediction for the sample without passing it through the rest of the backbone. At this point, the prediction can be sent to the procedure awaiting the results (e.g. an API, a socket connection, a callback). We shrink the batch by discarding the cache hits items at each exit and proceed with a smaller batch to the next (or the final) exit.

---

**Algorithm 1** Cache-enabled model inference

---

**Require:** Backbone                                           ▷ The original model
**Require:** CachedLayers                                 ▷ List of cached layers
**Require:** Layer           ▷ As part of Backbone, including the associated cache model and threshold
 1: **procedure** FORWARDPASS(X, callback)                           ▷ X: Input batch
 2:     **for Layer in Backbone.Layers do**                   ▷ In order of presence[1]
 3:         X ← Layer(X)

 4:         **if** Layer in CachedLayers **then**
 5:             Cache ← Layer.CacheModel
 6:             T ← Cache.Threshold
 7:             cachedPDs ← Cache(X)
 8:             confidences ← max(cachedPDs, axis=1)
 9:             callback(cachedPDs[confidences≥ T])              ▷ Resolve cache hits
10:             X ← X[confidences<T]                     ▷ Shrink the batch
11:         **end if**
12:     **end for**
13: **end procedure**

---

So far in the method, we have only evaluated the cache models individually, but to gain the highest improvement, we must also evaluate their collaborative performance within the cache-enabled model. Once the cache-enabled model is deployed, each cache model affects the following cache models' hit rates by narrowing the set of samples for which they will infer. More specifically, even if a cache model shows promising hit rate and accuracy in individual evaluation, its performance in the deployment can be affected due to the previous cache hits made by the earlier cache models (connected to shallower layers in the backbone). Therefore, we need to choose the optimum subset of cache models to infer the predictions with the minimum computations.

A brute force approach to find the optimum subset would require evaluating the cache-enabled model with each subset of the cache models. However, we implement a more efficient method without multiple executions of the cache-enabled model.

First, for each cache model, we record its prediction per sample in the $MD_L^{Val}$ and their confidence values. We also record two FLOPs counts per cache model; One is the cache model's FLOPs count($C_1$), and the other is the fallback FLOPs count which denotes the FLOPs in the remaining layers in the backbone($C_2$). For example, for the layer L12 in the Figure 1, $C_1$ is the corresponding cache model's FLOPs count, and $C_2$ is the FLOPs count in the layers L13 through L16.

For each subset $S$, we process the lists of predictions recorded for each model in $S$ to generate the lists of samples they actually receive when deployed along with other cache models in $S$. The processing consist of keeping only the samples in each list for which there has been no cache hits by the previous cache models in the subset. Further, we divide each list into two parts according to each cache model's confidence threshold; One consisting of the cache hits, and the other consisting of the cache misses.

Finally, we score each subset using the processed lists and recorded values for each cache model in $S$ as follows:

$$K(S) = \sum_{\Delta \in S} |H_\Delta| \times (C_{2,\Delta} - C_{1,\Delta}) - |M_\Delta| \times C_{1,\Delta} \tag{4}$$

Where

$K$     : The caching score for subset $S$
$\Delta$     : A cache model in $S$

---

[1]The loop is to show that each cache model will receive the cached layer's activation values when available, immediately, before proceeding to the next layer in the base model.

$H_\Delta$ : The generated list of cache hits for $\Delta$
$M_\Delta$ : The generated list of cache misses for $\Delta$
$C_{1,\Delta}$ : FLOPs count recorded for $\Delta$
$C_{2,\Delta}$ : Fallback FLOPs count recorded for $\Delta$

The score equation accounts for both the improvement a cache model provides through its cache hits within the subset, and the overhead it produces for its cache misses.

Final schemas after applying the method on MobileFaceNet, EfficientNet, ResNet18, and ResNet50 are discussed in the main text, with detailed illustrations provided in the Appendix (see Figure 6). This figure demonstrates the chosen subsets and their associated cache models for each backbone and dataset.

### 3.5 Early-Exit Optimization Implementation

In this section, we present the implementation and application of early-exit models for efficient and timely predictions in two distinct computer vision tasks: image classification and object detection. We begin by incorporating our proposed early-exit approach into architectures widely used for image classification tasks i.e. MobileFaceNet, EfficientNet, ResNet18, and ResNet50 on benchmark datasets i.e. CIFAR10, CIFAR100 and LFW.

Inspired by the promising results obtained in image classification, we further extend our methodology to address a critical real-world scenario: pedestrian detection in urban environments. For this purpose, we adopt the state-of-the-art Mask R-CNN model, renowned for its exceptional object detection capabilities. By integrating our early-exit strategy into Mask R-CNN, we enable the model to detect pedestrians at an earlier stage during inference, thus significantly reducing the processing time and providing timely warnings to autonomous vehicles about the presence of pedestrians within a given scene.

The significance of our contributions lies in the potential to enhance the safety and responsiveness of autonomous vehicles in urban settings, where pedestrian detection plays a pivotal role in avoiding accidents and ensuring seamless interaction between vehicles and pedestrians. In our approach, while prioritizing accuracy, we forgo an important aspect: the exact coordinates of the detected objects. By implementing early-exit models triggered upon the detection of pedestrians or humans, we aim to achieve faster processing and response times. However, we acknowledge that in certain cases, reacting within the required time window is of utmost importance. The trade-off between accuracy and response time is a crucial consideration in our methodology, and we recognize the significance of timely actions, especially in scenarios where immediate responses are critical for ensuring optimal outcomes.

In the context of the Mask R-CNN model, various options are available for selecting different backbones and settings, allowing for flexibility in performance evaluation and adaptation to specific tasks. While numerous configurations are possible, we opted to utilize a publicly available, pre-trained backbone to ensure that our experiments are standardized and well-established. This choice allows us to focus on the effectiveness of our proposed approach, leveraging the robustness and generalization capabilities of the chosen backbone. Additionally, using a pre-trained model helps to mitigate potential biases in training data and enables fair comparisons with other methods that have adopted similar backbones. The final schema of the cache models for Mask R-CNN with Resnet50 Backbone is illustrated in the Appendix (see Figure 7). We extend our early-exit classification model to implement object detection of pedestrians. The object detection model scans an input image and detects multiple objects within the image, assigning each detected object a probability distribution over possible classes.

In object detection, the model's task is identifying various objects within an image and providing a probability distribution for each detected object. The challenge therein lies in determining an effective method for updating the convolutional dense layer caches in this context.

### 3.5.1 Updating Caches for Pedestrian Detection

In object detection, pedestrians are one of the most frequently occurring classes. To optimize the performance of our early-exit framework for pedestrian detection, we explore three update strategies for the layers:

- Updating with the most confident pedestrians: In this approach, we selectively update the layers with features extracted from regions confidently classified as pedestrians. By focusing on the most confident detections, we aim to enhance the cache memory's relevance to crucial features associated with pedestrians in the scene.

- Updating with the most confident object: We investigate updating the layers with features from regions classified as the most confident object, regardless of whether it is a person or another class. This strategy is designed to ensure that the cache memory reflects critical features representative of the dominant object class in the scene.

- Updating with all detected objects: In this method, we update the layers with features from all detected objects in the scene. While this approach may provide a broader context, it may introduce redundancy and bias towards the more prevalent classes.

After testing these three cache-updating approaches, the results supported updating with the most confident single object as the best-performing method. Training cache layers with a sole focus on individual objects, such as pedestrians, leads to non-convergence and a lack of meaningful learning. Even prior to testing, it became evident that training layers exclusively with a defined class introduces bias, impeding effective learning. Meanwhile, training cache layers using a diverse set of objects results in model confusion, manifesting as reduced accuracy in our testing outcomes. So our selected approach updates the model with the most certain detection while avoiding the issues of bias and multi-object confusion.

### 3.6 Updates and maintenance

Similar to conventional caching, layer caching also requires recurring updates to the cache space to adapt to the trend in inference data. However, unlike conventional caching, we can not update the cache models in real-time. Therefore, to update the cache models using the extended set of collected inference samples, we retrain them and re-adjust their confidence thresholds.

The retraining adapts the cache models to the trend in the incoming queries and maintains their cache accuracy. We consider two triggers for the updates: I) When the size of the recently collected data reaches a threshold (e.g. 20% of the collected samples are new) and II) When the backbone is modified or retrained. However, users must adapt the recommended triggers to their requirements and budget.

## 4 Empirical Evaluation

In this section, we explain our experiment's objective, research questions, the tool implementation, and the experiment design including the backbones and datasets, evaluation metrics, and the environment configuration.

### 4.1 Objectives and research questions

The high-level objective of this experiment is to assess the ability of the automated layer caching mechanism to improve the compute requirements and inference time for DNN-based services.

To address the above objective, we designed the following research questions (RQ):

RQ1 To what extent the cache models can accurately predict the backbone's output and the ground truth data?
This RQ investigates the core idea of caching as a mechanism to estimate the final outputs earlier

in the model. The assessments in this RQ considers the cache models' accuracy in predicting the backbone's output (cache accuracy) and predicting the correct labels (GT accuracy).

RQ2 To what extent can cache-enabling improve compute requirements?
In this RQ, we are interested in how cache-enabling affects the models' computation requirements. In these measurements, we measure the FLOPs counts and memory usage as the metrics for the models' compute consumption.

RQ3 How much acceleration does cache-enabling provide on CPU/GPU?
In this RQ, we are interested in the actual amount of end-to-end speed up that a cache-enabled model can achieve. We break this result down to CPU and GPU accelerations, since they address different types of computation during the inference phase and thus may have been differently affected.

RQ4 How does the cache-enabled model's accuracy/latency trade-off compare with other early exit methods?
In this research question, we aim to assess and compare the performance of your cache-enabled model against other existing early exit methods concerning the trade-off between accuracy and latency in practical, real-world scenarios.

## 4.2 Tasks and datasets

Among the diverse set of classification tasks in real-world that are implemented by solutions utilizing DNN models, we have selected two representatives: face recognition and object classification. Both tasks are quite commonly addressed by DNNs and often used in large-scale services that have non-functional requirements such as: high throughput (due to the nature of the service and the large volume of input data) and are time-sensitive.

The face recognition models are originally trained on larger datasets such as MS-Celeb-1M (Guo et al., 2016) and are usually tested with different — and smaller — datasets such as LFW (Huang et al., 2008), CPLFW (Zheng et al., 2017), RFW (Wang et al., 2019), AgeDB30 (Moschoglou et al., 2017), and MegaFace (Kemelmacher-Shlizerman et al., 2016) for testing the models against specific challenges, such as age/ethnic biases, and recognizing mask covered faces.

We used the Labeled Faces in the Wild (LFW) dataset for face recognition which contains 13,233 images of 5,749 individuals. We used the images of 127 identities who have at least 11 images in the set so we can split them for training, validation and testing.

We also used CIFAR10 and CIFAR100 test sets (Krizhevsky, 2009) for object classification, each containing 10000 images distributed equally among 10 and 100 classes, respectively.

A reminder that we do not use the training data, rather we only use the test sets to simulate incoming queries at run-time. Specifically, we use only the test splits of the CIFAR datasets. However, we use the whole LFW data as it has not been used to train the face recognition models. Moreover, we do not use the labels in these test sets in the training and optimization process, rather we only use them in the evaluation step to provide GT accuracy statistics.

Each dataset mentioned above represents an inference workload for the models. Thus, we split each one into training, validation and test partitions with 50%, 20%, and 30% proportionality, respectively. However, we augmented the test sets using flips and rotations to improve the statistical significance of our testing measurements.

We employed the CityScape dataset to assess the presence of pedestrians ((Cordts et al., 2016)). This dataset is valuable for our research in urban scene comprehension, as it offers meticulously annotated images, with pixel-level labels, depicting various urban environments from the perspective of a vehicle. To evaluate the accuracy of our model, we needed labeled data for testing purposes. Upon observing that the test dataset within the CityScape dataset contained dummy labels, we opted to utilize the validation subset instead.

### 4.3 Backbones and Models

The proposed cache-enabling method is applicable to any deep classifier model. However, the results will vary for different models based on their complexity.

Among the available face recognition models, we have chosen well-known MobileFaceNet and EfficientNet models to evaluate the method, and we experiment with ResNet18 and ResNet50 for object classification.

The object classification models are typical classifier models out-of-the-box. However, the face recognition models are feature extractors that provide embedding vectors for each image based on the face/landmarks features. They can still be used to classify a face-identity dataset. Therefore, we attached a classifier block to those models and trained them (with the feature extractor layers frozen) to classify the images of the 127 identities with the highest number of images in the LFW dataset (above 10). It is important to note that since the added classifier block is a part of the pre-trained model under study, we discarded the data portion used to train the classifier block to ensure we still hold on to the constraint of working with pre-trained models without access to the original training dataset.

As previously stated, our pedestrian detection approach necessitated the selection of an object detection technique capable of identifying pedestrians within images. We adopted the Mask R-CNN framework. This method encompasses a backbone component (for which we employed ResNet50) and two additional sections that consume significant time and memory resources.

However, for our specific use case of providing early warnings to autonomous vehicles regarding the presence of pedestrians, the precise localization of pedestrians is not essential. Consequently, we chose to disregard the other resource-intensive sections, resulting in substantial time savings while still achieving the necessary level of awareness regarding pedestrian presence.

### 4.4 Metrics and measurements

Our evaluation metrics for RQ1 are ground truth (GT) accuracy and cache accuracy. Cache accuracy measures how accurately a cache model predicts the backbone's outputs (regardless of correctness). The GT accuracy applies to both cache-enabled model and each individual cache model. However, the cache accuracy only applies to the cache models.

In RQ2, we compare the original models and their cache-enabled version in terms of the average FLOPs count occurring for inference and their memory usage. We only measure the resources used in inference. Specifically, we exclude the training-specific layers (e.g., Batch Normalization and Dropout) and computations (e.g., gradient operations) in the analysis.

FLOPs count takes the model architecture and the input size into account and estimates the computations required by the model to infer for the input (Desislavov et al., 2021). In other words, the fewer FLOPs used for inference, the more efficient is the model in terms of compute and energy consumption.

On the other hand, we report two aspects of memory usage for the models. First is the total space used to load the models on the memory (i.e. model size). This metric is essentially agnostic to the performance of cache models and only considers the memory cost of loading them along with the backbone.

In addition to the memory required for their weights, DNNs also allocate a sizeable amount of temporary memory for buffers (also referred to as tensors) that correspond to intermediate results produced during the evaluation of the DNN's layers Levental (2022). Therefore, our second metric is the live tensor memory allocations (LTMA) during inference. LTMA measures the total memory allocated to load, move, and transform the input tensor through the model's layers to form the output tensor while executing the model.

In RQ3, we compare the average inference latency by the original model and its cache-enabled counterpart. Inference latency measures the time spent from passing the input to the model till it exits the model (by either an early-exit or the final classifier in the backbone). Various factors affect the inference latency including hardware-specific optimizations (e.g., asynchronous computation), framework, and model implementation. In our measurements, the framework and model implementations are fixed as discussed in Appendix .1. However, to account for other factors, we repeat each measurement for 100 times and report the average

inference latency recorded for each experiment. Further, to also account for the asynchronous computations effects in GPU inference latency, we repeated the experiments with different batch sizes.

Please refer to Appendix 5 for details of implementation and setup.

### 4.5 Experiment results

In this sub-section, we evaluate the results of applying the method on the baseline backbones and discuss the answers to the RQs. For a more comprehensive understanding, further details about our implementation, including access to the code repository, are provided in the Appendix 5.

#### 4.5.1 RQ1. How accurate are the cache models in predicting the backbone's output and the ground truth labels?

In this RQ we are interested in the built cache models' performance in terms of their hit rate, GT accuracy, and cache accuracy. We break down the measurements into two parts. The first part covers the cache models' individual performance over the whole test set without any other cache model involved. The second part covers their collaborative performance within in the cache-enabled model.

#### 4.5.2 Cache models' individual performance

Figure 2 portrays each cache model's individual performance against any confidence threshold value in CIFAR100-Resnet50 experiment. The figures demonstrating the same measurements for other experiments are available in Appendix .2.

We make three key observations here. First, deeper cache models are more confident and accurate in their predictions. For instance, cache 1 in the Figure 2 has 33.36% GT accuracy and 35.74% cache accuracy, while these metrics increase to 78.60% and 95.38% for Cache 3, respectively. This observation agrees with the generally acknowledged feature extraction pattern in the DNNs — deeper layers convey more detailed information.

The second key observation is the inverse correlation between the cache models' accuracy (both GT and cache) and their hit rates. This observation highlights the reliability of confidence thresholds in distinguishing the predictions more likely to be correct. For instance, cache 1 in Figure 2, with a 20% confidence threshold, yields 35.24% hit rate but also 8.99% drop in the final accuracy. However, with a 60% confidence threshold, it yields a 4% hit rate and does not reduce the final accuracy more than 0.1%.

The third observation is that the cache accuracy is higher than the GT accuracy in all cases. This difference is because we have trained the cache models to mimic the backbone only by observing its activation values in the intermediate layers and outputs. Since we have not assumed access to the GT labels (which is the case for inference data collected at run-time) while training the cache models, they have learned to make correct predictions only through predicting the backbone's output, which might have been incorrect in the first place. On the other hand, we observed that the cache models predict the correct labels for a portion of samples for which the backbone misclassifies. For instance, for 0.92% of the samples, cache 3 (in the Figure 2) correctly predicted the GT labels while the backbone failed ($\overline{B}C$ predictions). This shows the cache models' potential to partially compensate for their incorrect caches ($B\overline{C}$ predictions) by correcting the backbone's predictions for some samples ($\overline{B}C$). This indeed agrees with the overthinking concept in SDN (as discussed in 2.3) since for this set of samples, the cache models have been able to predict correctly in the shallower layers of the backbone.

#### 4.5.3 Cache models' collaborative performance

Table 2 describes the cache models' collaborative performance within the cache-enabled model per experiment. In the table, we also report how each cache model's cache hits have affected the final accuracy.

Here, we observe that while evaluating the cache models on the subset of samples, which were missed by the previous cache models (the relatively more complex ones), the measured hit rate and GT accuracy is substantially lower compared to the evaluation on the whole dataset. This is indeed due to the fact that the

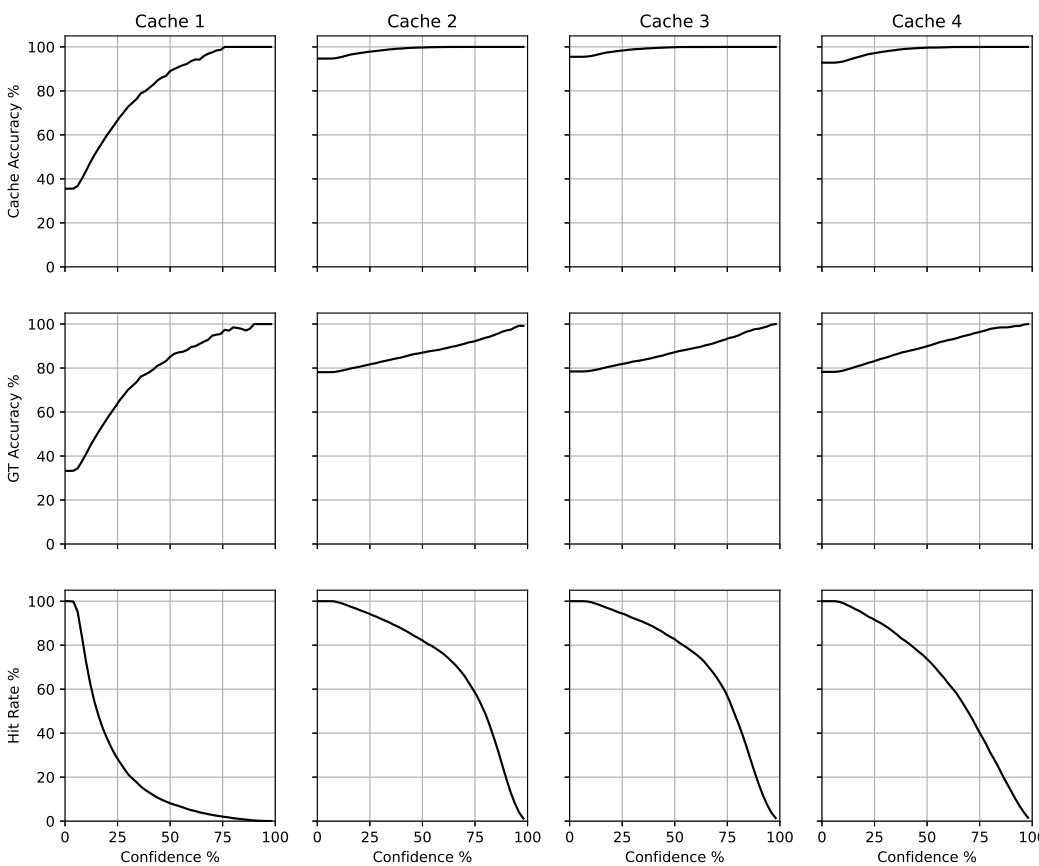

Figure 2: Individual accuracy and hit rate of the cache models vs. confidence threshold per cache model in CIFAR100 - Resnet50 experiment

Table 2: Cache models' collaborative performance in terms of hit rate(HR), cache accuracy ($A_{cache}$), GT accuracy ($A_{GT}$), and their effect on the final accuracy($\downarrow A_{effect}$). LFW: Labeled Faces in the Wild, MFN: MobileFaceNet, EFN: EfficientNet

| Data | Model | Final accuracy | | Exit# | HR | $A_{cache}$ | $A_{GT}$ | $\downarrow A_{effect}$ |
| | | Base | Cache-enabled | | | | | |
|---|---|---|---|---|---|---|---|---|
| CIFAR10 | Resnet18 | 88.71% | 86.49% | 1 | 67.21% | 92.29% | 88.91% | 01.31% |
| | | | | 2 | 10.33% | 89.76% | 76.63% | 0.56% |
| | | | | 3 | 11.24% | 85.71% | 51.43% | 0.25% |
| | | | | 4 | 8.32% | 91.37% | 35.71% | 0.1 % |
| | Resnet50 | 87.92% | 85.88% | 1 | 61.41% | 89.12% | 86.19% | 1.12% |
| | | | | 2 | 15.73% | 93.01% | 77.84% | 0.58% |
| | | | | 3 | 10.29% | 82.22% | 53.33% | 0.3% |
| | | | | 4 | 6.1% | 97.47% | 42.65% | 0.04% |
| CIFAR100 | Resnet18 | 75.92% | 74.47% | 1 | 11.96% | 99.29% | 82.11% | 0.94% |
| | | | | 2 | 58.26% | 99.62% | 85.41% | 0.1% |
| | | | | 3 | 7.26 % | 93.81% | 59.29% | 0.3% |
| | | | | 4 | 5.36% | 55.56% | 38.89% | 0.11% |
| | Resnet50 | 78.98% | 77.04% | 1 | 11.92% | 76.34% | 80.2% | 1.32% |
| | | | | 2 | 61.98% | 98.56% | 84.55% | 0.34% |
| | | | | 3 | 11.5% | 97.85% | 63.69% | 0.27% |
| | | | | 4 | 7.38% | 73.68% | 52.63% | 0.1% |
| LFW | MFN | 97.78% | 96.91% | 1 | 37.35% | 98.63% | 97.88% | 0.51% |
| | | | | 2 | 41.02% | 99.71% | 99.71% | 0% |
| | | | | 3 | 55.95% | 93.44% | 96.18% | 0.24% |
| | EFN | 97.29% | 95.35% | 1 | 63.73% | 96.82% | 96.24% | 1.67% |
| | | | | 2 | 14.52% | 99.12% | 98.76% | 0.02% |
| CityScape | Mask RCNN | 91.0% | 83.4% | 1 | 34.3% | 58.3% | 57.9% | 0.1% |
| | | | | 2 | 36.34% | 79.2% | 79.1% | 0.24% |
| | | | | 3 | 21.12% | 87.31% | 86.16% | 0.81% |

simpler samples (less detailed and easier to classify) are resolved earlier in the model. More specifically, hit rate decreases since the cache models are less confident in their prediction for the more complex samples, and GT accuracy also decreases since the backbone also is less accurate for such samples. However, we observe that the cache models still have high cache accuracy with low impact on the overall accuracy. This observation shows how the confidence-based caching method has effectively enabled the cache models to provide early predictions and keep the overall accuracy drop within the given tolerance.

### 4.5.4 RQ2. To what extent can cache-enabling improve compute requirements?

In this RQ, we showcase the amount of computation caching can save in terms of FLOPs count and analyze the memory usage of the models.

Table 3: Original and cache-enabled models FLOPs (M:Mega - $10^6$)

| Dataset(input size) | Model | FLOPs | | ↓ Ratio |
| | | Original | Cache-enabled | |
|---|---|---|---|---|
| CIFAR10($3 \times 32 \times 32$) | Resnet18 | 765M | 414M | 45.88% |
| | Resnet50 | 1303M | 601M | 53.87% |
| CIFAR100($3 \times 32 \times 32$) | Resnet18 | 766M | 374M | 51.17% |
| | Resnet50 | 1304M | 547M | 58.05% |
| LFW($3 \times 112 \times 112$) | MobileFaceNet | 474M | 296M | 37.55% |
| | EfficientNet | 272M | 182M | 33.08% |
| CityScape($3 \times 2048 \times 1024$) | Mask R-CNN | 4950M | 2730M | 44.84% |

Table 3 demonstrates the average amount of FLOPs computed for inference per sample. Here we observe that shrinking the batches proportionally decreases the FLOPs count required for inference.

Table 4: Original and cache-enabled models memory usage

| Dataset(input size) | Model | Original | | Cache-enabled | |
| | | Model Size | LTMA | Model Size | LTMA |
|---|---|---|---|---|---|
| CIFAR10($3 \times 32 \times 32$) | Resnet18 | 43MB | 102MB | 97MB | 88MB |
| | Resnet50 | 91MB | 235MB | 243MB | 201MB |
| CIFAR100($3 \times 32 \times 32$) | Resnet18 | 43MB | 104MB | 383MB | 93MB |
| | Resnet50 | 91MB | 235MB | 552MB | 189MB |
| LFW($3 \times 112 \times 112$) | MobileFaceNet | 286MB | 567MB | 350MB | 515MB |
| | EfficientNet | 147MB | 371MB | 297MB | 349MB |
| CityScape($3 \times 2048 \times 1024$) | Mask R-CNN | 3680MB | 3925MB | 4171MB | 4216MB |

Moreover, table 4 shows the memory used to load the models (i.e., the model size) and the total LTMA during inference while inferring for the test set. As expected, the cache-enabled models' size is larger than the original model in all cases since they include the backbone and the additional cache models. However, the decreased LTMA in all cases shows the reduced amount of memory allocations during the inference time. Generally, lower LTMA indicates smaller tensor dimensions (e.g. batch size, input and operators' dimensions) (Ren et al., 2021). However, in our case, since we do not change neither of the dimensions, lower LTMA is due to avoiding the computations in the remaining layers after cache hits which require further memory allocations. A noteworthy observation from this table highlights the substantial memory usage of our object detection approach due to the sizeable model employed. This underscores the notion that implementing caching for this purpose does not significantly amplify the memory requirements.

Although the FLOPs count and memory usage indicate the model's inference computational requirements, the decreased amount of FLOPs and LTMA does not necessarily lead to proportional reduction in the models' inference latency, which we further investigate in the next RQ.

#### 4.5.5 RQ3. How much acceleration does cache-enabling provide on CPU/GPU?

In this RQ, we investigate the end-to-end improvement that cache-enabling offers. The results of this measurement clearly depend on multiple deployment factors such as the underlying hardware and framework, and as we discuss later in the section, their asynchronous computation capabilities.

Table 5: end-to-end evaluation of cache-enabled models improvement in average inference latency, batch size = 32, MFN: MobileFaceNet, EFN: EfficientNet

| Dataset | Model | Original latency | | Cache-enabled latency | | ↓ Ratio | |
|---|---|---|---|---|---|---|---|
| | | CPU | GPU | CPU | GPU | CPU | GPU |
| CIFAR10 | Resnet18 | 13.4 ms | 1.08 ms | 10.11 ms | 0.98 ms | 24.55% | 10.2% |
| | Resnet50 | 18.73 ms | 1.81 ms | 14.62 ms | 1.51 ms | 31.08% | 16.57% |
| CIFAR100 | Resnet18 | 14.23 ms | 1.39 ms | 9.39 ms | 1.25 ms | 34.01% | 10.08% |
| | Resnet50 | 19.59 ms | 2.05 ms | 9.02 ms | 1.84 ms | **46.08%** | 16.75% |
| LFW | MFN | 25.34 ms | 8.22 ms | 16.91 ms | 7.30 ms | 33.23% | 11.19% |
| | EFN | 39.41 ms | 17.63 ms | 27.98 ms | 14.38 ms | 29.01% | 18.44% |
| CityScape | Mask R-CNN | 895 ms | 145.2 ms | 562.3 ms | 108.7 ms | 45.12% | **35.32**% |

Table (5) shows the average latency for the base models on CPU and GPU, vs. their cache-enabled counterparts, evaluated on the test set.

The first key observation here is the improvements on CPU. This improvement is due to the low parallelism in the CPU architecture. Essentially, the computations volume on CPU is proportional to the number of samples. Therefore, when a sample takes an early-exit, the remaining computation required to finish the tasks for the batch proportionally decreases.

$\boxed{\text{Review 1-5}}$ **[[ The FLOPs of all our cache models, utilized by each input in the batch, are aggregated, resulting in latency being dictated by the longest time observed in the batch. Consequently, a larger batch size tends to worsen latency, as it may lead to more frequent cache misses. ]]** The second observation is the relatively lower latency improvement on GPU. This observation shows that shrinking a batch does not proportionally reduce the inference time on GPU, which is due to the high parallelism in the hardware. Shrinking the batch on GPU provides a certain overhead since it interrupts the on-chip parallelism and hardware optimizations. This interruption forces the hardware to re-plan its computations which can be time consuming. Thus, batch shrinking improvements can be insignificant on GPU. The third observation pertains to time savings related to pedestrian detection, as distinct from the primary model. This significant gain in efficiency is attributed to disregarding the additional layers of Mask R-CNN through our early exit strategy.

Table 6: Inference latency improvement on GPU vs. batch size in Resnet18 and Resnet50 trained on CIFAR100

| Model | Batch Size | Original Latency | Cache-enabled Latency | ↓ Ratio |
|---|---|---|---|---|
| Resnet18 | 16 | 1.34 ms | 1.18 ms | 11.83% |
| | 32 | 1.39 ms | 1.25 ms | 10.08% |
| | 64 | 1.43 ms | 1.77 ms | -24.28% |
| | 128 | 1.61 ms | 2.11 ms | -31.05% |
| Resnet50 | 16 | 1.98 ms | 1.71 ms | 13.68% |
| | 32 | 2.05 ms | 1.84 ms | 16.75% |
| | 64 | 2.19 ms | 1.98 ms | 9.21% |
| | 128 | 2.7 ms | 3.22 ms | -19.43% |

Table 6 further demonstrates how the batch size affects the improvement provided by caching. The key observation here is that increasing the batch size can negate the caching effect on the inference latency

which as discussed is due to fewer number of batches that are fully resolved through the cache models and do not reach the last layers. In conclusion, the latency improvement here highly depends on the hardware used in inference and must be specifically analyzed per hardware environment and computation parameters such as batch size. However, the method still can be useful when the model is not performing batch inferences (batch size = 1). One can also use the tool and get a best prediction so far within the forward-pass process by disabling the batch shrinking. Doing so will generate multiple predictions per input sample, one per exit (early and final).

### 4.5.6 RQ4. How does the cache-enabled model's accuracy/latency trade-off compare with other early exit methods?

In this RQ, we conducted a comprehensive evaluation of three distinct methods, including BranchyNet(Teerapittayanon et al., 2016), Gati(Balasubramanian et al., 2021), and our proposed method on a same system configuration. This evaluation included two different Resnets with two different datasets to assess the performance of each method thoroughly.

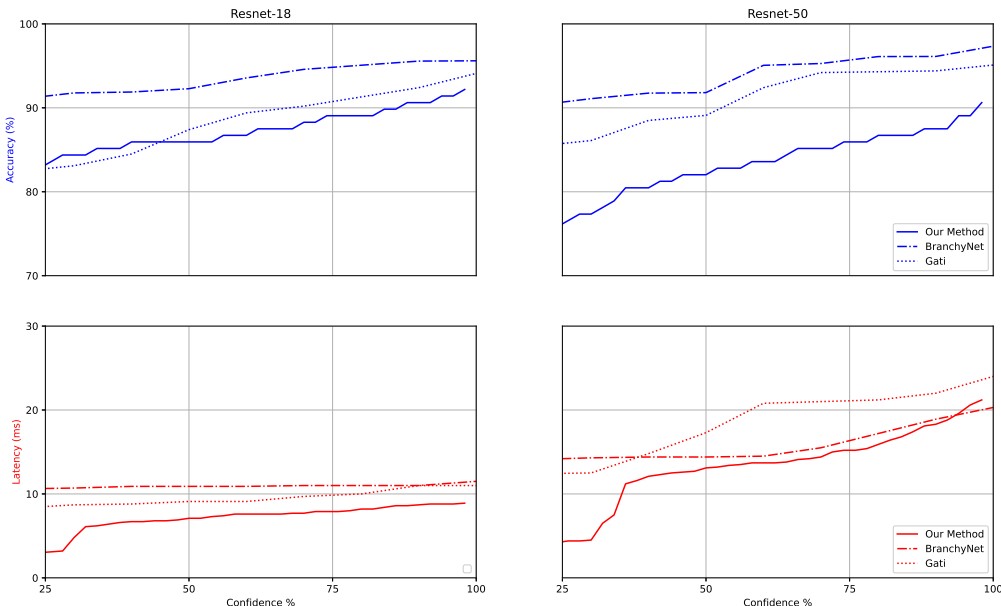

Figure 3: CIFAR10 Accuracy/latency comparison between the approaches with different confidence.

Figures 3 and 4 demonstrate comprehensive analysis of the accuracy/latency trade-off across various confidence levels for all evaluated methods for CIFAR10 and CIFAR100. Our findings reveal a remarkable adaptability of our proposed method to different confidence levels, particularly excelling in low-confidence scenarios. This adaptability underscores the effectiveness of our approach's training, showcasing its ability to perform exceptionally well in situations where traditional methods might falter. Conversely, as confidence levels increase, our method exhibits a slight latency increase, reflecting its ability to finely tune confidence settings to suit various application requirements. This feature demonstrates the method's adaptability and its potential for accommodating diverse use cases.

The primary observation is that our method exhibits the best latencies across various methods at different confidence levels. Another significant observation is that our method's performance scales positively with the complexity of the model or dataset, leading to superior outcomes in terms of both accuracy and latency. As we augment some lightweight layers for each model, it becomes evident that Resnet50 exhibits more pronounced improvements in our research. This observation is also depicted in Figure 3. For the simplest

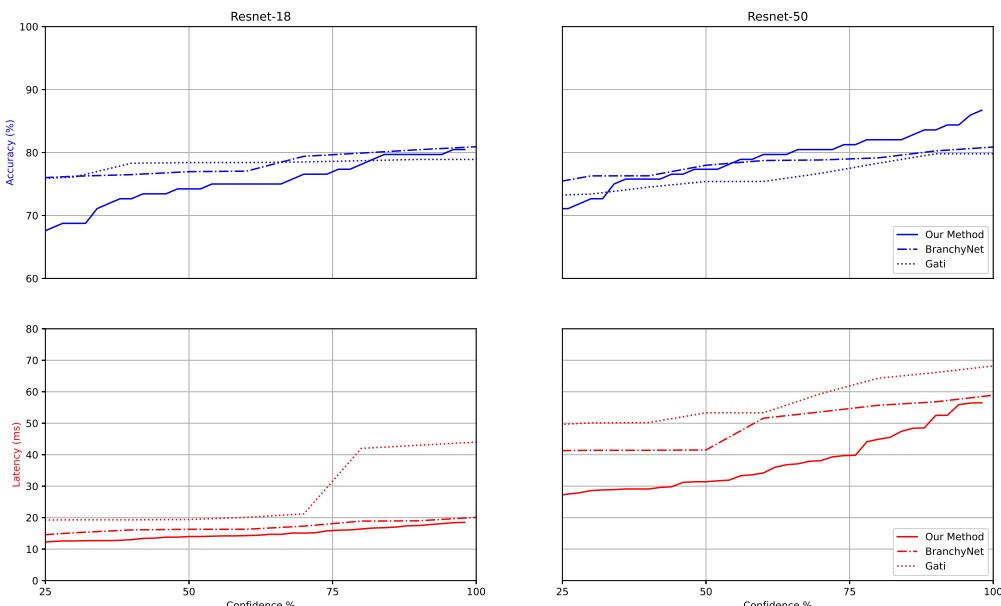

Figure 4: CIFAR100 Accuracy/latency comparison between the approaches with different confidence.

model and dataset, namely CIFAR10-Resnet18, we achieved a substantial 52.2% and 32.4% reduction in latencies, with a modest 6.1% and 0.9% decrease in mean accuracy when compared to the BranchyNet and Gati methods, respectively. In the case of a more complex model and dataset, such as CIFAR100-Resnet50, we achieved a substantial 51.6% and 30.4% reduction in latency while simultaneously improving mean accuracy by 2.92% and 0.87% when compared to the Gati and BranchyNet methods, respectively. BranchyNet, on the other hand, exhibits relatively consistent results across different confidence levels. While this stability might be desirable in some scenarios, it lacks the adaptability and dynamic response that our method offers. Gati, while showing learned caches, is marked by the complexity of its training process, which can be challenging to implement effectively. Moreover, Gati consumes more memory resources during implementation, contrasting with the efficiency of our approach.

Indeed, our method stands out as a dynamic and adaptable solution, requiring less extensive data and model preparation compared to approaches like Gati. The ability to configure confidence levels further underscores its versatility for a broad spectrum of applications. It is noteworthy, however, that for simpler models and datasets, our method may not demonstrate a marked improvement over others, as they employ additional dense layers for caching which can provide competitive results. Crucially, the most significant aspect of our work lies in the capability to update the cache layers effectively without reliance on ground truth labels, offering a substantial advantage in practical applications where such labels may not be readily available.

### 4.6 Limitation and future directions

The first limitation of this study is that the proposed method is limited to classification models since it would be more complicated for the cache models to predict a regression model's output due to their continuous values. This limitation is strongly tied to the effectiveness of knowledge distillation in case of regression models.

The method also does not take the internal state of the backbone (if any) into account, such as the hidden states in recurrent neural networks. Therefore, the method's effectiveness for such models still needs to be further assessed.

Moreover, practitioners should take the underlying hardware and the backbone structure into account as they directly affect the final performance. On this note, as shown in section 4.5.5, different models provide different performances in terms of inference latency in the first place, therefore, choosing the right model for the task comes first, and caching can be helpful in improving the performance.

While our current comparison has yielded valuable results, we can explore the applicability of our approach to other large models, particularly in non-vision-based datasets, to assess its effectiveness in different domains. Given the growing importance of reducing latency and inference time, especially in Large Language Models (LLMs), our future research can focus on methods to further optimize and reduce costs for large-scale industries."

## 5    Conclusion

In this paper, we have showed that our automated caching approach is able to extend a pre-trained classification DNN to a cache-enabled version using a relatively small and unlabelled dataset. The required training dataset for caching models are collected just by recording the input items and their corresponding backbone outputs at the inference time. We have also shown that the caching method can introduce significant improvement in the model's computing requirements and inference latency, specially when the inference is performed on CPU.

We discussed the parameters, design choices, and the procedure of cache-enabling a pre-trained off-the-shelf model, and the required updates and maintenance.

In conclusion, while traditional caching might not be beneficial for DNN models due to the diversity, size and dimensions of the inputs, caching the features in the hidden layers of the DNNs using the cache models can achieve significant improvement in the model's inference computational complexity and latency. As shown in sections 4.5.4 and 4.5.5, caching reduces the average inference FLOPs by up to 58% and the latency up to 46.09% on CPU and 18.44% on GPU for classification purposes. For pedestrian detection we could reduce latency up to 45.1% on CPU and 35.32% on GPU. In summary, our method consistently outperforms alternative approaches across different models and datasets. In the case of the simplest model, CIFAR10-Resnet50, we observe a remarkable 52.2% and 32.4% reduction in latency, with accuracy only experiencing a minor 6.1% and 0.9% decrease compared to BranchyNet and Gati methods, respectively. For the more intricate CIFAR100-Resnet50 model and dataset, our method excels with a significant 51.6% and 30.4% reduction in latency, and a notable 2.92% and 0.87% improvement in mean accuracy compared to Gati and BranchyNet methods. These findings underline the adaptability and superior performance of our method in diverse and complex scenarios.

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

## Appendix A. Implementation

### .1 Implementation details

We developed the caching tool using PyTorch, which is accessible through the GitHub repository[2]. Figure 5 shows the overall system design. The tool provides a NAS module, an optimizer module, and a deployment

---

[2]https://anonymous.4open.science/r/AutoCacheLayer-CBB4/

module. The NAS module provides the architectures to be used per cache model. The optimizer assigns the confidence thresholds, finds the best subset of the cache models and provides evaluation reports. Lastly, the deployment module launches a web server with the cache-enabled model ready to serve queries.

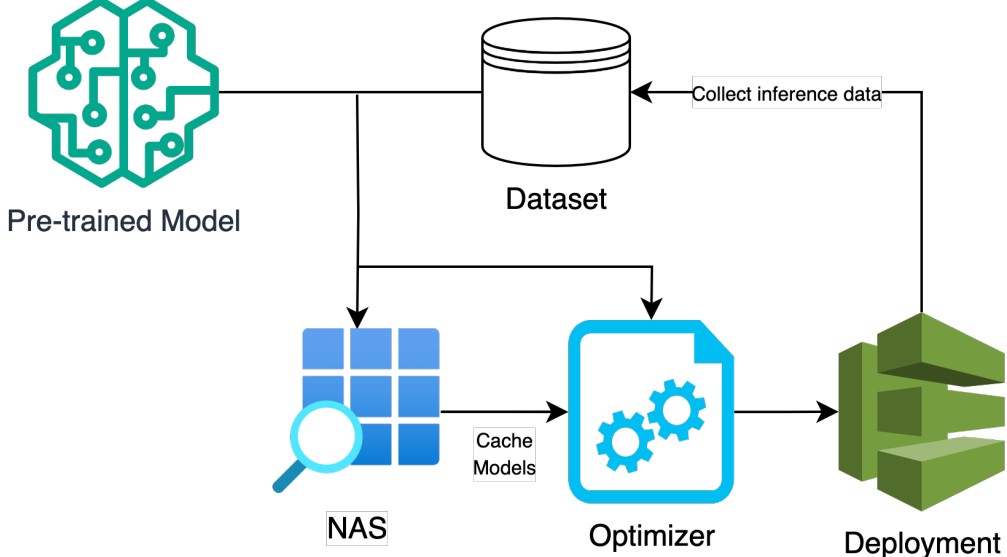

Figure 5: Caching system overall framework

### .1.1   NAS Module

Existing NAS tools typically define different search spaces according to different tasks which constrains their applicability to certain input types and sizes. Using such tools with input constraints defeats our method's generalization and automation purpose since the cache models' inputs can have any dimension and size. For instance, ProxylessNAS (Cai et al., 2019) specializes in optimizing the neural architecture performance for a target hardware. However, it is only applicable for image classification tasks and requires certain input specifications (e.g., 3xHxW images normalized using given values). Similarly, Auto-PyTorch (Zimmer et al., 2021) and Auto-Keras are only applicable to tabular, text, and image datasets.

We chose NNI by Microsoft (Microsoft, 2022) as it does not constrain the model inputs in terms of type, size, and dimensions. NNI also provides an extensible search space definition with support for variable number of layers and nested choices (e.g., choosing among different layer types, each with different layer-specific parameters).

Given the backbone implementation, the dataset, and the search space, the module launches an NNI experiment per candidate layer to search for an optimum cache model for the layer. Each experiment launches a web GUI for the progress reports and the results.

We aim for end-to-end automation in the tool. However, currently, the user still needs to manually export the architecture specifications when using the NAS module and convert them to a proper python implementation (i.e., a PyTorch module implementing the architecture). The specifications are available to the user through the experiments web GUI and also in the trial output files. This shortcoming is due to the NNI implementation, which does not currently provide access to the model objects within the experiments. We have created an enhancement suggestion on the NNI repository to support the model object access (issue #4910).

### .1.2   Optimizer and deployment modules

Given the backbone's implementation and the cache models, the optimizer evaluates cache models, assigns their confidence thresholds, finds the best subset of the cache models and disables the rest, and finally reports

the relevant performance metrics for the cache-enabled model and each cache model. We used the DeepSpeed by Microsoft and PyTorch profiler to profile the FLOPs counts, memory usage, and latency values for the cache models and the backbones.

The user can use each module independently. Specifically, the user can skip the architecture search via the NAS module and provide the architectures manually to the optimizer, and the module trains them before proceeding to the evaluation.

The tool also offers an extensive set of configurations. More specifically, the user can configure the tool to use one device (e.g., GPU) for training processes and the other (e.g., CPU) for evaluation and deployment.

The deployment module launches a web server and exposes a WebSocket API to the cache-enabled model. The query batches passed to the socket will receive one response per item, as soon as the prediction is available through either of the (early or final) exits.

### .1.3   Backbone Implementation

We used the backbone implementations and weights provided by the FaceX-Zoo (Wang et al., 2021) repository to conduct the experiments with LWF dataset on MobileFaceNet and EfficientNet models.

For experimenting with CIFAR10 and CIFAR100, we used the implementations provided by torchvision (Marcel & Rodriguez, 2010) and the weights provided by (Phan, 2021) and (Weiaicunzai, 2020).

All the backbone implementations were modified to implement an interface that handles the interactions with the cache models, controls the exits (cache hits and misses), and provides the relevant reports and metrics. We documented the interface usage in the repository, so users can experiment with new backbones and datasets. We refer interested readers to a blog post on how to extract intermediate activations in PyTorch (Bhaskhar, 2020) which introduces three methods to access the activation values. The introduced forward hooks method in PyTorch is very convenient for read-only purposes. However, our method requires performing actions based on the activation values, specifically, cache lookup and batch shrinking and avoiding further computation through the next layers. Therefore, we used the so called "hacker" method to access the activation values and perform these action and provided the interface for easy replication on different backbones.

### .2   Environment setup

The hardware used for inference substantially affects the results due to the hardware-specific optimizations such as computation parallelism. In our experiments, we have used an "Intel(R) Core(TM) i7-10700K CPU @ 3.80GH" to measure on-CPU inference times and an "NVIDIA GeForce RTX 3070" GPU to measure on-GPU inference time.

## Appendix B. Final schema of the cache models

This appendix shows two key figures that provide a deeper visual understanding of the methodologies and results highlighted in the main text.

Figure 6 illustrates the final schema of the cache models used in our experiments with CIFAR10-Resnet18, CIFAR100-Resnet50, LWF-EfficientNet, and LFW-MobileFaceNet.

Figure 7 presents the final schema of the cache models specifically designed for our experiments with CitiScapes using the Mask R-CNN framework.

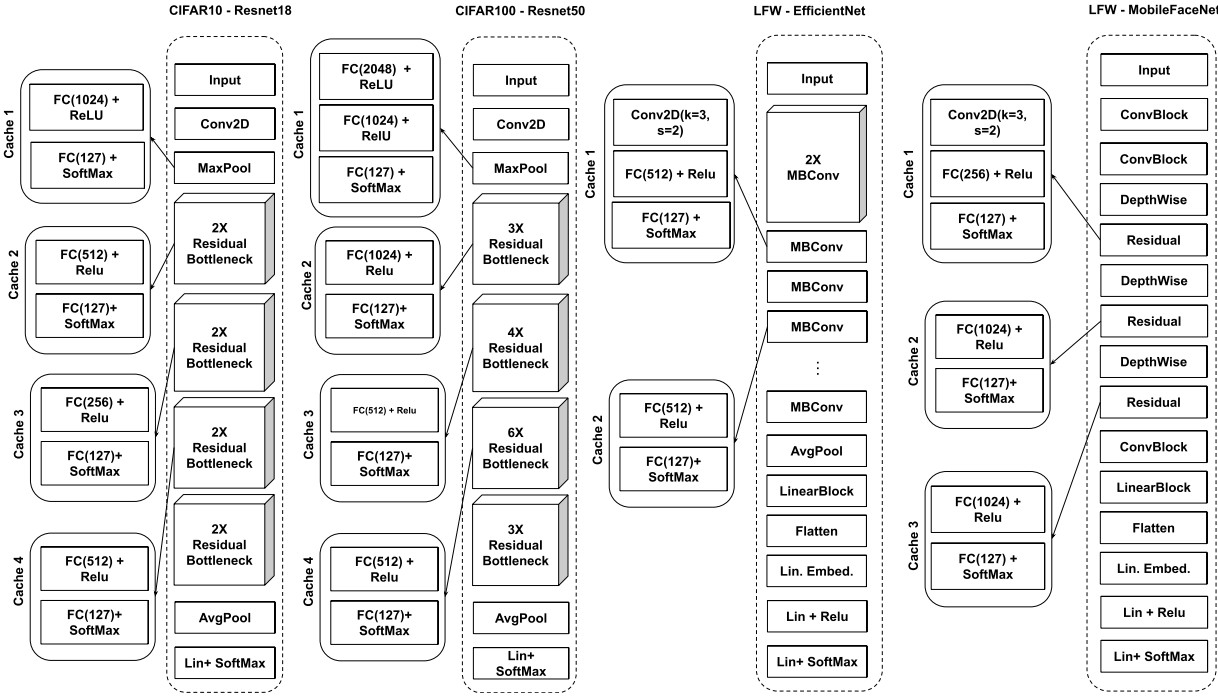

Figure 6: Final schema of the cache models, for the experiments CIFAR10-Resnet18, CIFAR100-Resnet150, LWF-EfficientNet, and LFW-MobileFaceNet

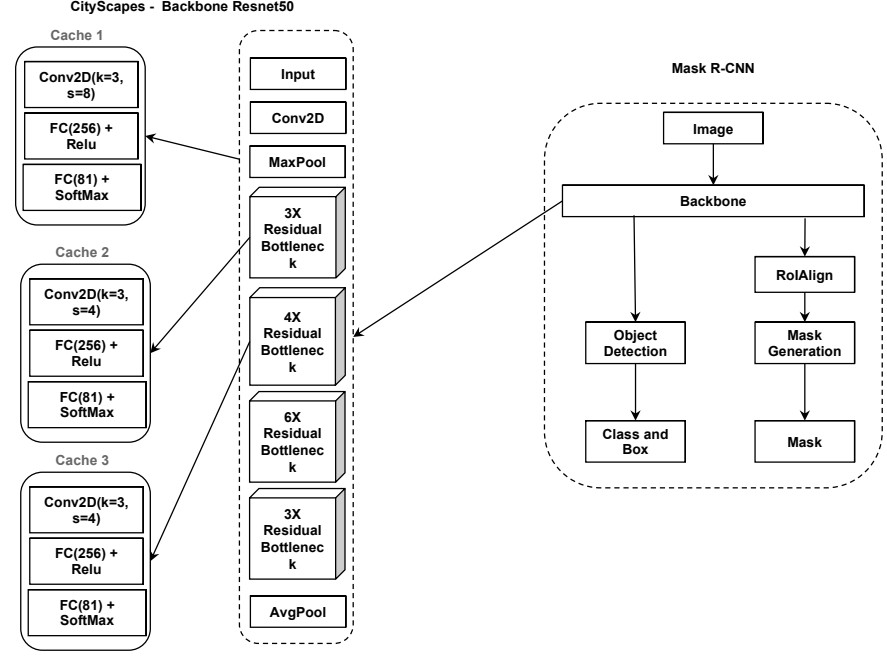

Figure 7: Final schema of the cache models, for the experiments CitiScapes - Mask RCNN

## Appendix C. Cache models' individual performance for all experimenst

The following figures demonstrate the hit rate, GT accuracy, and cache accuracy of each cache model vs. the confidence threshold, per experiment dataset and backbone.

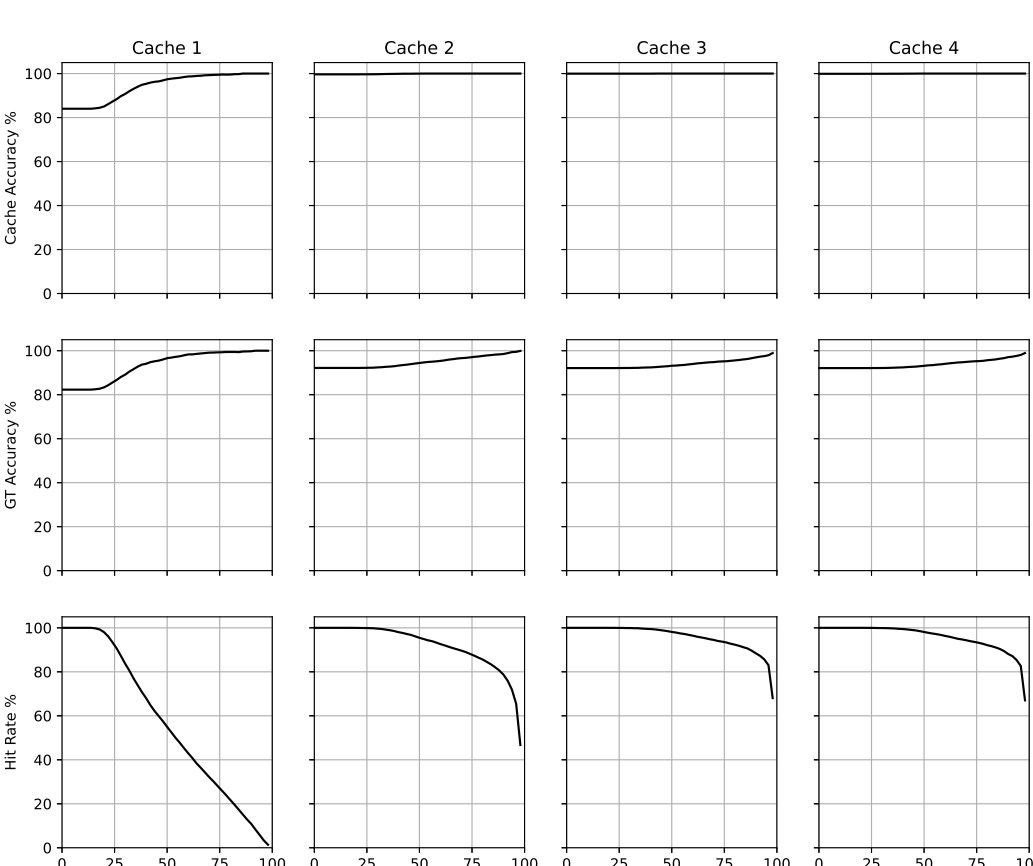

Figure 8: Experiment: CIFAR10-Resnet18

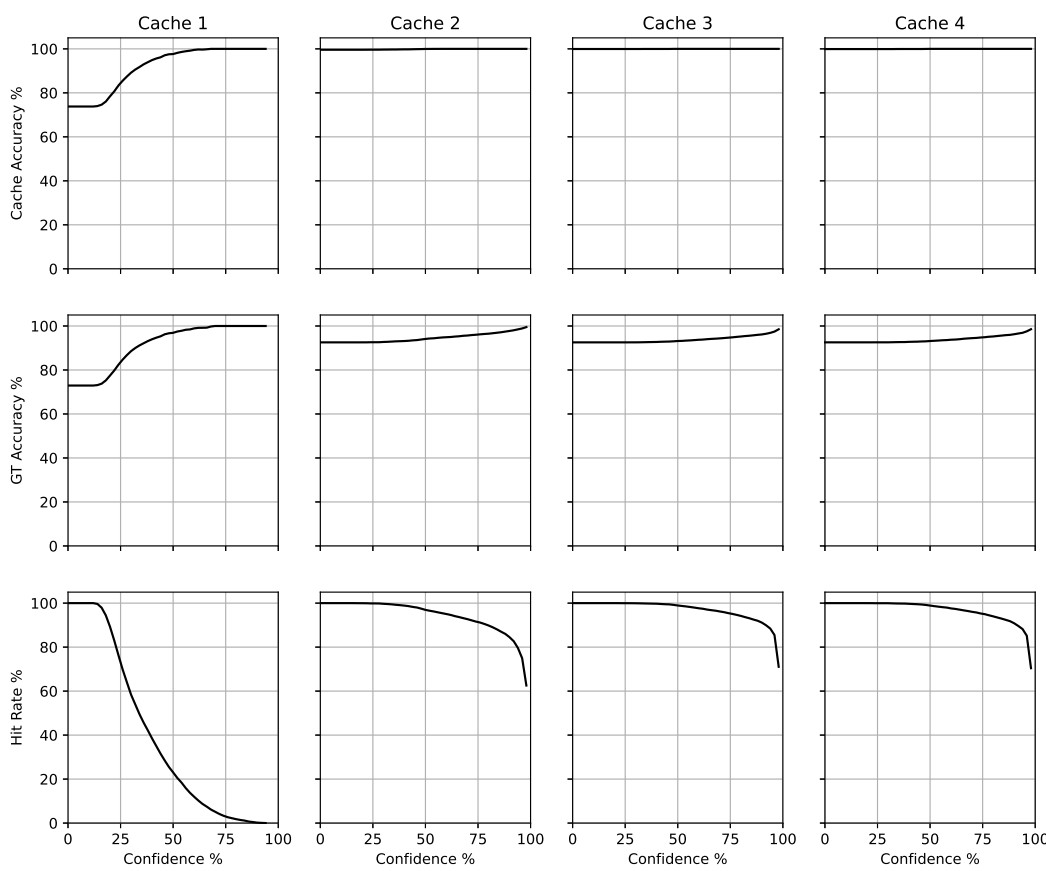

Figure 9: Experiment: CIFAR10-Resnet50

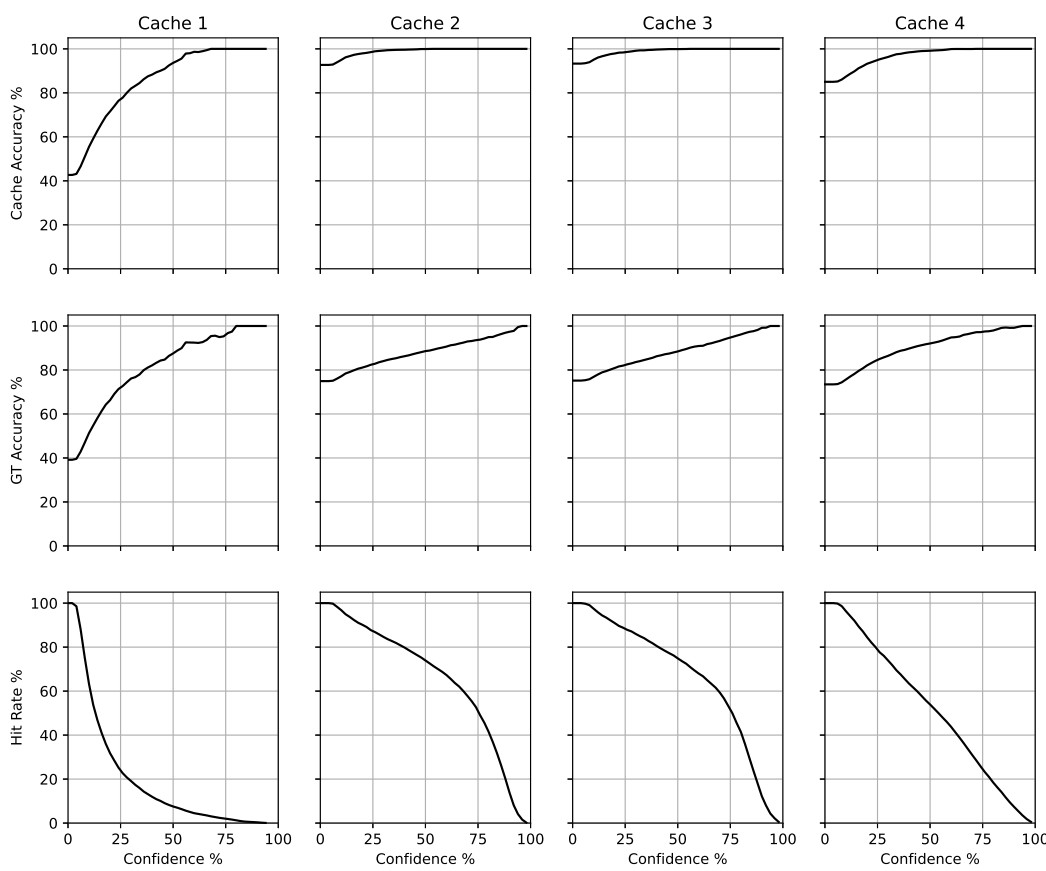

Figure 10: Experiment: CIFAR100-Resnet18

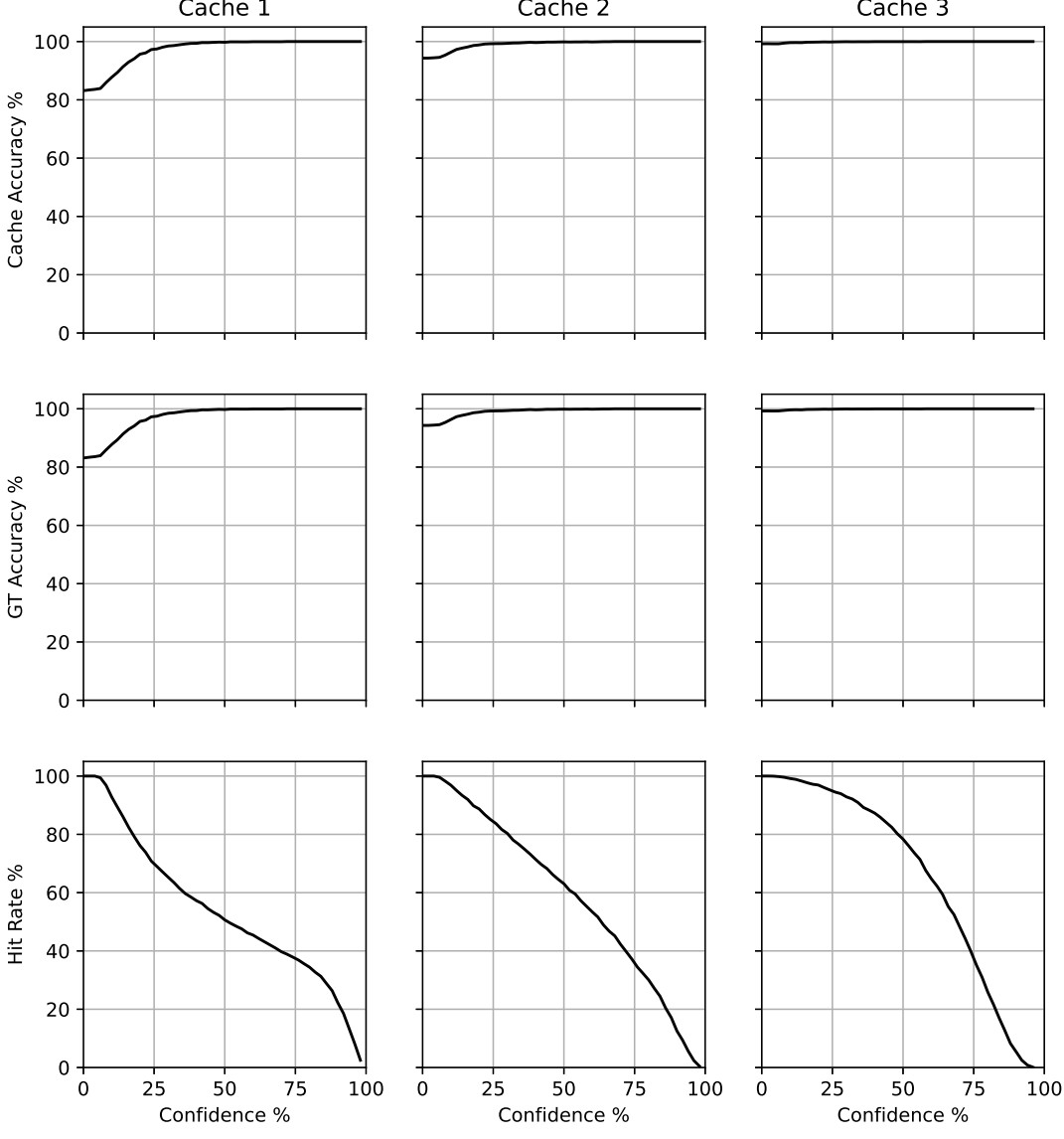

Figure 11: Experiment: LFW-MobileFaceNet

# LFW-EfficientNet

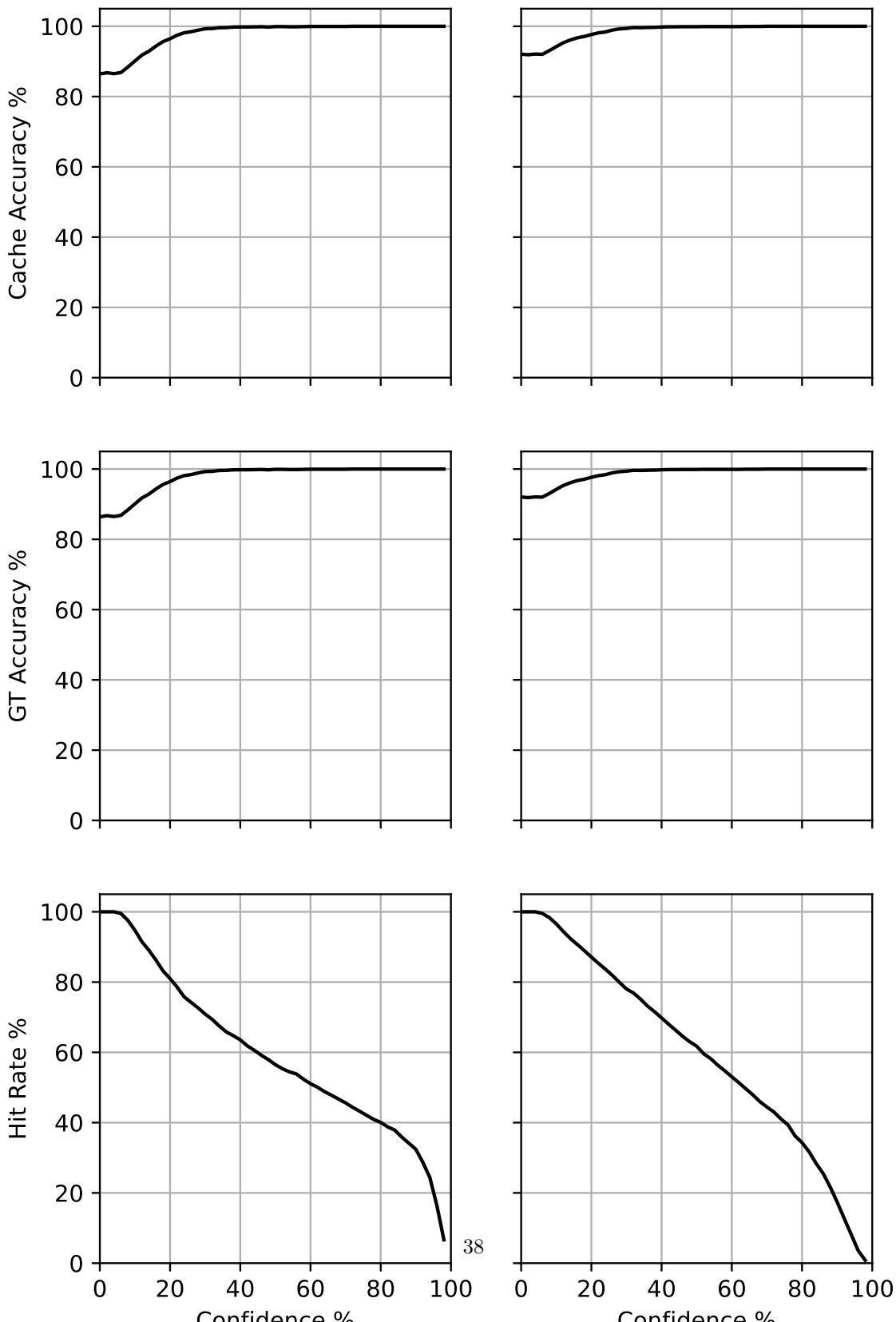

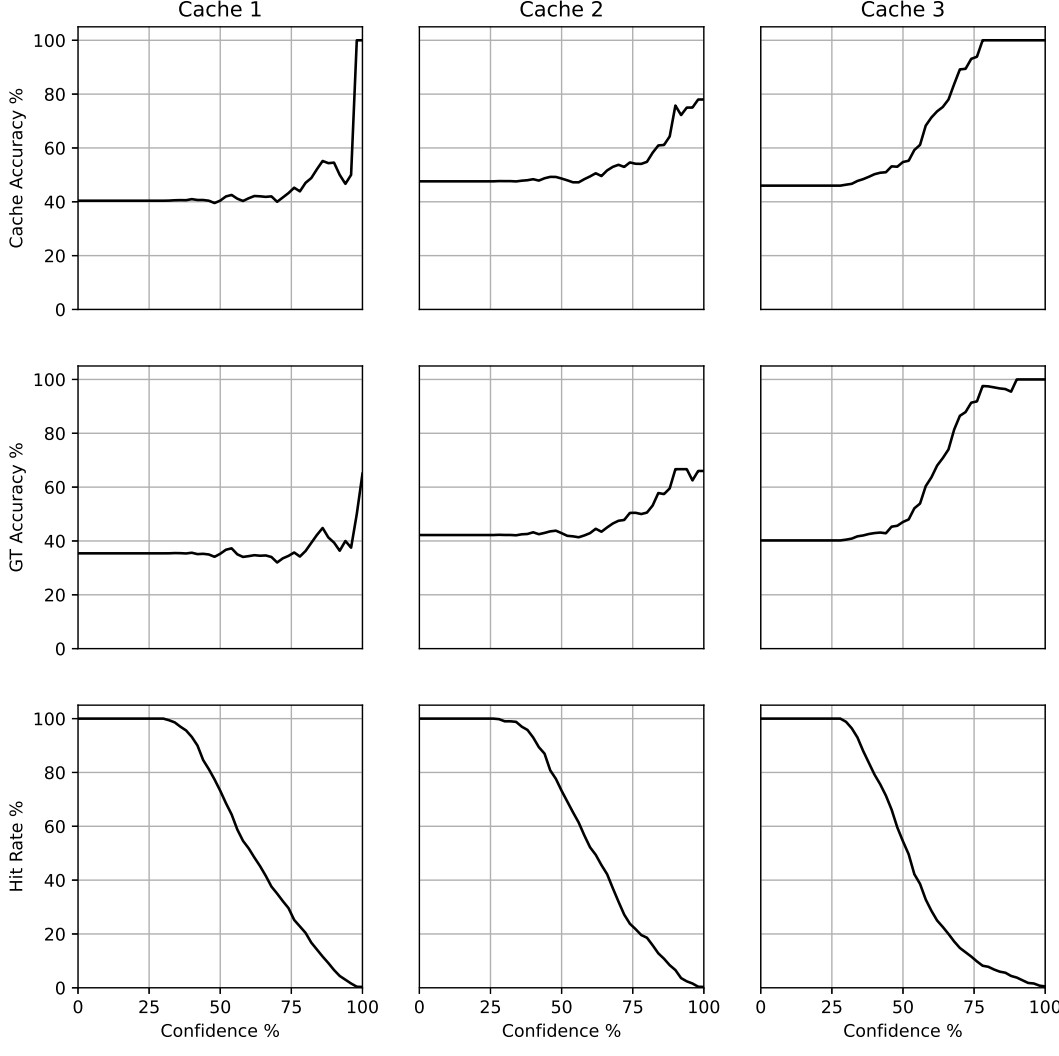

Figure 13: Experiment: Mask RCNN-CityScape

