# OpenReview forum: "Improving Efficiency of Neural Image Classification and Object Detection Systems using Automated Layer Caching"
_TMLR — Rejected by TMLR_

### Review · Reviewer_P9yW · 2023-12-24

**Summary Of Contributions:**

The authors propose an automated caching framework solution to help improve the computational and latency of a deep neural network (DNN). They derive ideas from self-distillation and early-exit methodologies.

The main idea is to utilize network architecture search methodologies to first identify cache model architectures at relevant layers in the DNN. These cache models are then trained separately using KL divergence (similar to self-distillation techniques) and assigned a confidence threshold. These cache models are then scored on the validation set, to select the optimum set of cached models.

The authors claim the following benefits of their framework:
- Reliance only on unlabeled inference data (in contrast to existing works that use labelled inference data)

- Reduced latency/computational overhead

- Enhanced batch processing capabilities due to the usage of batch shrinking.

**Audience:**

Yes

**Claims And Evidence:**

Yes

**Requested Changes:**

**Minor Changes:**

- Typographical changes – The authors should go through the paper to correct the typographical errors made.  For e.g., SOTA is misspelt as STOA in the introduction, the abbreviation LFW is utilized before the expansion in section 4.2.

- Clarifications in the choice of wording – The authors should explain what “providing dominating results” means on page 2.

**Major Changes:** The authors should address the points mentioned in the “weaknesses” section.

**Strengths And Weaknesses:**

**Strengths:**
- The paper is fairly well written (barring a few typographical errors) and easy to follow.

- The framework proposed addresses the important issue of reducing computational and latency overhead of DNNs.

- Not relying on labelled data has practical advantages.

**Weaknesses:**

The paper could be strengthened by addressing the following weaknesses:

-  This work has similarities to [1]. The authors in [1] present a framework that is capable of handling unlabeled data (self-distillation) similar to the proposed framework in addition to handling labelled data. It would be beneficial to cite and compare the self-distillation case of this work.

- Providing a comparison of latency/computational cost of other techniques mentioned in the related works (such as inference optimization) would strengthen the novelty of the proposed framework.

- The authors present a use case of this framework in the real-world scenario of pedestrian detection for autonomous vehicles. However, they trade off response time of the network to accuracy of pedestrian detection. When utilizing any detection scheme in autonomous vehicles, the accuracy and exact location of the objects are as important as the response time. In other words, a false but quickly made decision does not provide any advantage. The authors should consider demonstrating the value of the proposed framework using other applications, ones where a drop in accuracy can be tolerated, and even preferred for gaining computational/latency benefits.

- The proposed framework trains cache models separately (i.e., freezes the rest of the network including other cached models while training). The optimum subset of these cached models (that work hand in hand) are then selected using a scoring mechanism. How does this work in comparison to training cached models together? Providing this information could help the reader understand the need for (and even appreciate) the proposed training technique.

- The search space for the  cache models have been restricted to 2 convolution layers/2 fully connected layers (for the author’s experiments). Would increasing this space by allowing deeper (3 to 5 layers) cache models allow early exits at earlier layers? If so, would this be computationally less expensive than the shallower cached models exiting at later layers of the backbone? An ablation study exploring these questions would be a good addition to the paper.

- How are the cache-enabled FLOPs and cache-enabled latency time calculated in section 4.5.5 when utilizing larger batch sizes? Are the FLOPs of all cached models utilized by each of the inputs in the batch aggregated? Is the latency the largest time for any input in the batch? The authors should explain this.

- How does the proposed framework compare in terms of the runtime memory requirements? Making a comparison of this factor in addition to the latency and computational comparisons would provide a more wholistic (and a stronger) picture of the framework.

*[1] Leontiadis, Ilias, et al. "It's always personal: Using early exits for efficient on-device CNN personalisation." Proceedings of the 22nd International Workshop on Mobile Computing Systems and Applications. 2021.*

---

> ### Author Response · Authors · 2024-02-09
> **General Responses to Reviewer P9yW**
>
> Dear Reviewer,
>
> Thank you for the fast review process and the constructive comments and suggestions. We have carefully considered all comments and have planned to address them:
>
> **Comment 1:This work has similarities to [1]. The authors in [1] present a framework that is capable of handling unlabeled data (self-distillation) similar to the proposed framework in addition to handling labeled data. It would be beneficial to cite and compare the self-distillation case of this work.
> [1] It's always personal: Using early exits for efficient on-device CNN personalisation.**
>
> Thanks for suggesting this paper. We have thoroughly reviewed it and will cite and discuss the differences in the paper. In short, in this paper, they try to train their cache models **using labels** and as they have mentioned, it is **semi-supervised**. But we train the layers using only the trained models. Not using any labels during training the caches is our main contribution compared to existing work.
>
> **Comment 2:Providing a comparison of latency/computational cost of other techniques mentioned in the related works (such as inference optimization) would strengthen the novelty of the proposed framework.**
>
> Thanks for the suggestion. In the revised version we will add latency/ computational cost results for the baseline techniques, on Resnets and CIFARs. Note that object detection can not be implemented using the related works, since they are optimized and developed only for classification methods.
>
> **Comment 3:The authors should consider demonstrating the value of the proposed framework using other applications, ones where a drop in accuracy can be tolerated, and even preferred for gaining computational/latency benefits.**
>
> Good suggestion. Please note that even with our current case studies the drop in accuracy is minimal (less than 1% in the best cases). Nonetheless, we plan to add another case study of DLRM (Deep Learning Recommendation Model) on Criteo Ad Kaggle dataset. We selected this case study for these reasons: (a) unlike the previous case we had in the paper, the recomm systems in practice can tolerate accuracy decrease to achieve lower latency and higher response rate, (b) the dataset is much larger than the two we used to have, (c) the task is completely new so it helps in verifying the generalizability of the findings.
>
> **Comment 4:The optimum subset of these cached models (that work hand in hand) are then selected using a scoring mechanism. How does this work in comparison to training cached models together?**
>
> To find the best subset to use as our cache models and layers we benefit from the NAS approach. It does not guarantee to select the best model (it is reasonable because it only  performs on limited models and data without complete training), we have designed an ablation study which can test both our methods and this scoring as well (Please see the next comment regarding the details about the experiments).
>
> **Comment 5:The search space for the cache models have been restricted to 2 convolution layers/2 fully connected layers (for the author’s experiments). Would increasing this space by allowing deeper (3 to 5 layers) cache models allow early exits at earlier layers? If so, would this be computationally less expensive than the shallower cached models exiting at later layers of the backbone? An ablation study exploring these questions would be a good addition to the paper.**
>
> In the revised version we will add an ablation study as follows:
> We will focus on the design of cache layers and study three different configurations: (a) using NAS we will find the top N solutions and we see how the results are different from each other (b) without NAS, we pick one random cache configuration and run it on all models and show how much decrease of performance we get compare to the NAS-optimized approach, and (c) we also do a small sensitively analysis to see to what extent the choice of cache layers’ numbers/type matter.
>
>
> **Comment 6:How are the cache-enabled FLOPs and cache-enabled latency time calculated in section 4.5.5 when utilizing larger batch sizes? Are the FLOPs of all cached models utilized by each of the inputs in the batch aggregated? Is the latency the largest time for any input in the batch?**
>
> Yes, the FLOPs of all our cached models, utilized by each input in the batch, are aggregated. Yes, latency is the largest time seen in the batch. We will elaborate on this in our revised manuscript.
>
> **Comment 7:How does the proposed framework compare in terms of the runtime memory requirements? Making a comparison of this factor in addition to the latency and computational comparisons would provide ...**
>
> Section 4.5.2 already explains the details of run-time memory consumption, but given your comment we will provide further details on memory consumption in comparison to other baseline in RQ4.
>
> **Comment 8:Some typographical changes.**
>
> We make sure we will proof-read the revised version again more carefully.

---

### Review · Reviewer_uRWL · 2024-01-02

**Summary Of Contributions:**

This paper proposes an approach to automated layer caching in deep neural networks for image classification and object detection models. The core ideas of the proposed method are the use of self-distillation of DNN models and early-exist. The proposed method was empirically evaluated with two types of tasks and four datasets.

**Audience:**

Yes

**Claims And Evidence:**

Yes

**Requested Changes:**

- The revision should better clarify the novel contributions of the proposed approach and contrast it to the existing work.
Sec. 3 needs rigorous explanations to justify the design of each of the steps in the proposed approach.

- Ablation studies regarding the alternative choices for each component are suggested. The authors should at least provide better justifications for their design in Sec. 3.

- Sec. 4.5 is not well-organized. I would personally suggest having an individual section, namely empirical evaluation results. Since the authors have explicitly defined four research questions, I would suggest nesting results and findings under each research question.

- Currently, Sec. 4.5.2 and 4.5.3 are in awkward positions. Finally, using one or two sentences to summarize the answers to each research question may improve the paper’s readability.

**Strengths And Weaknesses:**

### Strengths
- A timely and important topic
- This paper is well-written
- Great performance of the proposed method.

### Weaknesses
- Limited novelty and research insights
- Lack of rigor in the design of the proposed approach

### Detailed comments
Overall, this paper targets an important topic. The overall presentation is good in terms of explaining the details of the proposed approach. According to the empirical evaluation results, the proposed method also achieved SOTA performance.

However, my major concern comes from the limited novelty and research insights. The proposed approach seems to be an ad-hoc combination of several existing approaches (e.g., distillation, early-exist). In this way, I did not see rich novel contributions.

I also found the design of the proposed approach somehow lacks rigor. For instance, the criteria listed in Sec 3.1. seems to come from nowhere. There are no literature references or empirical studies supporting the design choices.

In terms of the evaluation, there is no ablation study in its current form. However, given that the proposed method includes multiple components ( e.g., distillation, which should have alternative choices), ablation studies are needed to mitigate the threats to validity.

---

> ### Author Response · Authors · 2024-02-09
> **General Responses to Reviewer uRWL**
>
> Dear Reviewer,
>
> Thank you for the fast review process and the constructive comments and suggestions. We have carefully considered all of your comments and have planned to provide answers or address them:
>
> **Comment 1:The revision should better clarify the novel contributions of the proposed approach and contrast it to the existing work.**
>
> Our work’s main novelty is about training the cache models only at inference time and without any labels. We will make this distinction with related work more explicit in the paper.
>
> **Sec. 3 needs rigorous explanations to justify the design of each of the steps in the proposed approach.**
>
> In a nutshell, in the first step, we analyze each selected model to find the candidate positions for cache layers. Layers with the outputs without any other layer’s state involved are the candidates which can enable training possibility.
> Then using NAS, we evaluate all possible subsets we can put in cache layers. NAS would also score between architectures we have defined for our cache models.
> Next we train the identified optimal layers at inference time without any labels.
> Finally, we test our cache models..
> We make sure these steps are better explained in the revised version.
>
> **Comment 2:Ablation studies regarding the alternative choices for each component are suggested. The authors should at least provide better justifications for their design in Sec. 3.**
>
> We plan to do an ablation study as suggested by the other reviewer. The design is explained in our response to that. We will also better explain each design choice that is not part of the ablation study, in the revised paper.
>
> **Comment 3:Sec. 4.5 is not well-organized. I would personally suggest having an individual section, namely empirical evaluation results. Since the authors have explicitly defined four research questions, I would suggest nesting results and findings under each research question.**
>
> Thanks for the suggestion. In the revised version we make two separate sections as 4. Empirical Evaluation Design and 5. Empirical Evaluation Results, where each RQ will have its own subsection.
>
> **Comment 4:Currently, Sec. 4.5.2 and 4.5.3 are in awkward positions. Finally, using one or two sentences to summarize the answers to each research question may improve the paper’s readability.**
>
> Thanks for the suggestion. We will add summary boxes per research question.

---

### Review · Reviewer_qwkm · 2024-01-24

**Summary Of Contributions:**

In this paper, the authors proposed to improve the efficiency of deep neural networks by attaching multiple classifiers to the intermediate layers. The attached classifiers are termed as cache models in this work. The cache models are searched by neural architecture search methods. Confidence threshold are selected to determine cache hits and misses. An evaluation metric is proposed to select the optimal subset of cache models. Experiments are done for detection and classification tasks to answer four research questions.

**Audience:**

Yes

**Broader Impact Concerns:**

I do not see ethical issues that need to be dealt with for this work. The only issue is that in the experiment human-face dataset is used. Those datasets are from public sources.

**Claims And Evidence:**

Yes

**Requested Changes:**

1. Why is NAS necessary and important? Why using the same architecture for the cache model would not work?
2. Are the architecture searching and training of the cache model conducted simultaneously?
3. Please explain in detail the following:
More specifically, even if a cache model shows promising hit rate and accuracy in individual evaluation, its performance in the deployment can be affected due to the previous cache hits made by the earlier cache models (connected to shallower layers in the backbone)
4. Page 7: medial dataest -> medial dataset

**Strengths And Weaknesses:**

Strength
1. The idea of using multiple exits for pre-trained models is quite interesting.
2. The experimental results show that the efficiency of models could improved somehow.

Weakness
1. The paper is not well written, the writing should be improved. A lot of details are not well explained.
2. After reading Section 3.3, it is still difficult to understand how confidence calibration is done. It is easy to understand what confidence calibration is. But it is not explained why the confidence of a trained cache model could be improved by using the validation set. This part is quite vague.
3. In the last but second paragraph of Sec. 3.3, notations are not explained well. What does $T$ and $X/2^n$ mean? Why is the accuracy drop decreases with the increase of the number of layers.
4. In Section 3.4, it is not clear how are the subsets $S$ decided. Once the subsets are generated, is it true that the subset with the best score in Equation 4 is selected?
5. It is better that the Sec 4.4 and Sec 4.5 are merged. The explanation of the metrics occurs where it is used for the first time.
6. Table 3 and Table 4 should be combined with the metric of accuracy drop. Considering the FLOPs or memory usage solely without mentioning accuracy drop makes no sense.
7. The reduction of computation is not significant compared with network compression methods (Table 2) [1,2]. In addition, the model size is increased and the memory usage could also be increased.
8. Experiments are not done on large scale datasets such as ImageNet

[1] Liu, Zechun, et al. "Metapruning: Meta learning for automatic neural network channel pruning." Proceedings of the IEEE/CVF international conference on computer vision. 2019.

[2] Li, Yawei, et al. "Dhp: Differentiable meta pruning via hypernetworks." Computer Vision–ECCV 2020: 16th European Conference, Glasgow, UK, August 23–28, 2020, Proceedings, Part VIII 16. Springer International Publishing, 2020.

---

> ### Author Response · Authors · 2024-02-09
> **General Responses to Reviewer qwkm**
>
> Dear Reviewer,
>
> Thank you for the fast review process and the constructive comments and suggestions. We have carefully considered all comments and have planned to address them:
>
> **Comment 2:After reading Section 3.3, it is still ...**
>
> In short, we employ a threshold to gauge the confidence level of the cache based on its model's output. Each cache functions as a classifier, and its output, during validation or testing, carries a confidence value. We will make further efforts to clarify this aspect in our revision.
>
> **Comment 3:In the last but second paragraph of Sec. 3.3 ...**
>
> It seems like we had a typo and X is wrongly mentioned in the paper and should be T which is the tolerance for drop in the final accuracy. n is the 1-based index of the cache model in the setup.
> NAS tries to minimize the total accuracy no more than the tolerance so for all possible subsets, there will be a maximum range. This method does not guarantee the best score but is a good solution. In the ablation study,  this will be explored as well.
>
> Regarding the second question, utilizing additional cache layers may cause earlier detection of a class with a higher probability (with using an appropriate confidence for cache layers) before moving to the next layers, so the total accuracy would increase with the cost of memory consumption.
>
> **Comment 4:In Section 3.4, it is not clear how ...**
>
> Exactly. After running our algorithm, we will select the optimal subset for use in our cache layers. We will add this in the revision:
> “Subsets with the best score will be selected as our cache system which will be added inside the model to be trained individually by the predictions and perform early exits.”
>
> **Comment 5:It is better that the Sec 4.4 and Sec 4.5 are merged ...**
>
> Based on the other reviewer’s comments we plan to create two separate sections of 4. Empirical Evaluation Design and 5. Empirical Evaluation Results, where each RQ will have its own subsection.
> Given this reorganization, we believe Sec 4.4 which explains metrics fall under design not results, and should go with Section 4 not 5.
>
> **Comment 6:Table 3 and Table 4 should be combined with the metric of accuracy drop ...**
>
> You are correct. We may add accuracy drops to the tables as well to make the comparison easier for the readers.
>
> **Comment 7:The reduction of computation is not significant ...**
>
> In **MetaPruning** and **DHP** mentioned in the comment, the authors try to use some search algorithms for channel pruning and they suggest compressed networks with lighter models which lead to a lower accuracy but faster inference. The computation cost would be really better than ours, in some certain cases. But our approach does not lose as much accuracy as them in most of the time, while showing faster inference. In addition, in our approach, computation can be tuned with the confidence parameter, which would be another advantage for us. We must mention that in most cases they offer better computation, but by having less accuracy or better accuracy with higher latencies.
> We will ensure to highlight these observations more clearly in our paper.
>
> Regarding the second part of the comment: Indeed, in certain scenarios, adding more cache layers and consequently increasing memory usage does not lead to significant improvements, but we demonstrate better accuracy using cache methods.
>
> **Comment 8:Experiments are not done on large scale datasets such as ImageNet.**
>
> Yes, our classifications primarily utilize simpler datasets. However, we acknowledge the Cityscapes (which contains high-resolution images) and have demonstrated results for it accordingly.
> We will perform tests on the Criteo Ad Kaggle dataset to show another application of this work and it will cover this up.
>
> **Comment 9:Why is NAS necessary ...**
>
> The initial assumption was that NAS is used to select the optimal architecture. We plan to run an ablation study to verify or reject this hypothesis, in the revised version.
>
> **Comment 10:Are the architecture searching and ...**
>
> No! We initially select the best model through NAS, and subsequently, we conduct our experiments and cache training in the next stage. We will make it more clear in the revised paper.
>
> **Comment 11:Please explain in detail the following...**
>
> We employ cache layers in series, meaning if a cache layer performs poorly in the early stages of our network and triggers an incorrect early exit, then a superior cache model positioned in the later stages might not be utilized. Consequently, this scenario could negatively impact the overall performance. For example, assume that we conduct an image for inference and a cache layer hits it soon with a wrong prediction. Then later layers will not happen to observe the input to decide about another early exit and the whole performance will decrease.
> So, the individual cache model's accuracy/hit rate don't guarantee the overall good results since the final results depend on the earlier cache model's results as well.

---

### Comment · Action_Editor_ugT3 · 2024-02-19
**Please Post a Revision, If Desired, to Inform the Reviewer Recommendations**

Dear Authors,

Reviewers have mentioned that they have read the responses, but would appreciate a revision of the paper, so they can better consider the changes. The TMLR process allows for revising the submission PDF and continued discussion, so please feel free to do so although there is no requirement.

Allow me to suggest a deadline of Feb. 26 (anywhere on earth) to give a full week's time.

Best regards,
Your Action Editor

---

### Decision · Action_Editor_ugT3 · 2024-03-21

**Recommendation:** Reject

**Comment:**

The feedback from reviewers is mixed while their recommendations are overall negative: two of the three reviewers vote for rejection ("Reject" and "Leaning Reject") while the remaining reviewer votes for "Leaning Accept" but does not champion the submission. The authors responded to the reviews and posted a single revision, but mention a plan for multiple revisions. Nevertheless the recommendations by the reviewers and the decision by the editor must be made on the actual revision and only estimates of the planned revisions. To summarize, there are points for and against both dimensions of claims/evidence and audience, but reviewers emphasize a need to clarify and to justify claims by further comparison (for instance: grounding the point about latency/computational cost by comparing with baselines) for acceptance.

At this time it is appropriate to reject the submission, in line with the reviewer recommendations, while allowing for revision and resubmission. Since the authors have signaled they plan further revisions, the action editor encourages them to make these revisions and do so.

Note: As clarification of the TMLR process, the nominal timeline for revisions is during the author-reviewer discussion phase, which is the two weeks following the return of all assigned reviews. While this process can be extended, the ~2 month scope for further changes—expressed by the authors in their revision—makes this case more suitable for resubmission.

**Audience:**

All reviewers agree positively on the existence of an audience. P9yW: [this work] addresses the important issue of reducing computational and latency overhead of DNNs. uRWL: timely and important topic. qwkm: using multiple exits for pre-trained models is quite interesting. The AE agrees that the topic is in scope for readers of TMLR, as the computational efficiency of deep network inference is of broad interest, and the content of this work is relevant because it provides technical material and experimental results.

However, reviewers P9yW and qwkm comment in their recommendations that the present submission may not reach its audience because "the writing of this paper should be further improved" and while "many questions were raised by [myself and the other reviewers]" only some questions were addressed by the response and revision while others remain pending because "[the authors] plan to address numerous clarifications and make substantial revisions". Furthermore feedback is provided (by P9yW, uRWL) on the need to expand the scope (more applications) and scale of experiments (such as ImageNet) to more broadly and better inform the TMLR audience.

The audience dimension is therefore borderline given these issues and the intermediate state of the submission.

**Claims And Evidence:**

All reviewers agree positively on the claims and evidence in their ratings, but disagree in their comments and overall summarization of their recommendations. On the positive side, strengths include the online and unsupervised nature of the caching, which can be done at test time without labels, the reported improvements in efficiency vs. inference without the technique, and the lack of a drop in accuracy for efficiency improvements. On the negative side, the key concerns are missing comparisons or discussions to justify claims of efficiency (such as latency/computational cost (P9yW), evaluation of network compression as an alternative (uRWL), and computational metrics are not jointly reported with accuracies (qwkm)) and the lack of ablations to understand the combination of parts that make up the method (O9yW, qwkm, uRWL; Point 4 in the author response in the revision). These concerns are partly addressed, but largely not present in the revision, and rather are promised for future revisions.

Factors regarding novelty plus the choice and scale of experiments were discussed, but these either do not or only partially impact assessment of claims of evidence:

- While reviewers asked for further results that could strengthen the paper (P9yW: "other applications, ones where a drop in accuracy can be tolerated" and uRWL: "large scale datasets such as ImageNet"), these are not strictly necessary or an obstacle to the claims and evidence, because such results are not claimed in this work.
- While reviewers raised novelty concerns (P9yW: "this work is [similar] to (Leontiadis et al., 2021) [and it would be] beneficial to cite and compare" and uRWL: "should better clarify the novel contributions of the proposed approach and contrast it to the existing work") the action editor recalls that novelty is not a decision criterion of TMLR. However, it is important that novelty claims are inaccurate, even if the degree of novelty is not emphasized by this venue. In this case revision is needed to more clearly identify the stated novelty of pure inference time and unsupervised caching w.r.t. existing work.

**Resubmission Of Major Revision:**

The authors may consider submitting a major revision at a later time.